# A general pharmacodynamic interaction model identifies perpetrators and victims in drug interactions

Sebastian G. Wicha[1], Chunli Chen[1], Oskar Clewe[1] & Ulrika S.H. Simonsson[1]

Assessment of pharmacodynamic (PD) drug interactions is a cornerstone of the development of combination drug therapies. To guide this venture, we derive a general pharmacodynamic interaction (GPDI) model for ≥2 interacting drugs that is compatible with common additivity criteria. We propose a PD interaction to be quantifiable as multidirectional shifts in drug efficacy or potency and explicate the drugs' role as victim, perpetrator or even both at the same time. We evaluate the GPDI model against conventional approaches in a data set of 200 combination experiments in *Saccharomyces cerevisiae*: 22% interact additively, a minority of the interactions (11%) are bidirectional antagonistic or synergistic, whereas the majority (67%) are monodirectional, i.e., asymmetric with distinct perpetrators and victims, which is not classifiable by conventional methods. The GPDI model excellently reflects the observed interaction data, and hence represents an attractive approach for quantitative assessment of novel combination therapies along the drug development process.

[1] Department of Pharmaceutical Biosciences, Uppsala University, Uppsala 75124, Sweden. Correspondence and requests for materials should be addressed to S.G.W. (email: sebastian.wicha@uni-hamburg.de)

Combination therapies are prevalent in many therapeutic areas[1], such as infectious diseases, oncology or neurology. There is a strong focus in drug development programs to early identify, quantify and evaluate pharmacokinetic (PK) interactions. However, for pharmacodynamic (PD) interactions, although at least equally important as PK, there is substantially less focus. Reasons for neglecting PD interactions may lay in the very heterogeneous methodological landscape[2], statistical traps[3,4], but also a general difficulty to adapt current PD interaction assessment methods to different stages of drug development.

Moreover, limitations of the current methodology landscape to evaluate PD interactions are numerous: (i) graphical approaches such as the isobologram method[5], fractional inhibitory concentration (FIC) indices[6] or the combination index[7] are conceptually straightforward and useful, but their results are difficult to interpret when interactions are concentration dependent or isoboles are "curvilinear"[8], e.g., when a combination partner is a partial agonist. (ii) Response surface approaches[9] are a frequently employed in such situations as they can elucidate concentration dependencies in the interaction space but, as outlined above, cannot be used for calculating an interaction score or parameter such as FIC indices. As response surface analyses represent a pure comparison between observed and additive response, they cannot be used for longitudinal simulations of the observed interaction pattern at (changing) concentrations over time (PK–PD simulations). (iii) FIC indices, but also model-based approaches with a single interaction parameter[10,11] provide interaction scores for statistical interaction assessment and can be compared, but the single point estimate might not mirror the complexity of response surfaces; model-based approaches with more interaction parameters, e.g., polynomials to describe the interaction surface[12], might be more flexible to fit to the data, but their interaction polynomials are not interpretable. (iv) Most approaches are limited to two interacting drugs[2,10], (v) tied to a single underlying additivity concept, or even no established additivity criterion[12,13], and (vi) cannot be adapted to the various complexity of information obtained along the drug development process, i.e., reduced or more complex nested models of the same type can be applied. Finally, (vii) we aimed to explore the roles of each drug in PD interaction studies. Recent and current work is addressing these limitations, e.g., through more efficient factorial designs[14], innovative mechanism-independent statistical models[15], rescaling techniques to facilitate correct interaction scoring in two[16] and three drug combinations[17,18]. Moreover, the genetic impact on drug interactions is also increasingly studied[19] to assess the genetic robustness of PD interactions.

Yet, all these efforts could be complemented by a general pharmacodynamic interaction (GPDI) model overcoming all limitations (i–vii). In addition, the GPDI model should (a) provide quantitatively interpretable interaction point estimates, (b) not require knowledge on the precise mode of action, (c) be flexible enough to adapt to multi-drug combination data of various complexity, (d) enable compatibility with established additivity criteria such as Loewe Additivity[20] or Bliss Independence[21], (e) provide insight into perpetrators and victims in PD interaction networks, (f) offer time-course prediction to avoid confounders arising from analysis of sole endpoint data[22], and (g) allow for computer simulations for therapeutic profiling, bridging pre-clinical to clinical phase as well as clinical trial simulation.

In order to evaluate and compare the performance and interaction classification of the here newly derived GPDI approach against conventional methods, we utilize a large-scale high-throughput screening data set[23]. Isobole analysis[5,23] and the Greco model[24] as methods derived from Loewe Additivity, as well as an empiric Bliss Independence model[23] are used as conventional approaches and compared to the result of the GPDI analyses.

## Results

**The general pharmacodynamic interaction model.** Single drug effects were characterized with the sigmoidal maximal effect (Emax) model[25,26], parameterized by PD parameters Emax (maximal drug effect), EC50 (drug concentration stimulating 50% of Emax, i.e., drug potency) and $H$ (Hill factor for sigmoidicity). The concept of the GPDI approach is simple: we propose a PD interaction to be quantifiable as shift in Emax (allosteric type) or EC50 (competitive type), which provides an intuitive, mechanistically motivated, quantitative, and statistically interpretable (point) estimate of a PD interaction. A central aspect to the GPDI model is its ability to define perpetrators and victims of a PD interaction: a perpetrator alters the PD parameter of the victim drug leading to a PD interaction, i.e., either synergy or antagonism. The interactions in the GPDI approach are directionally quantified, i.e., the drug can take the role of perpetrator, victim, or even both at the same time. This definition of perpetrator and victim is to our best knowledge new in the context of PD interactions, yet similar to the use of such terms in the context of PK interactions, where, e.g., drug elimination of a victim drug is reduced by a perpetrator drug[27]. To include PD interactions, we extended the sigmoidal Emax model of the victim drug by a perpetrator sigmoidal Emax term ("GPDI term") to capture the interaction effect on the level of the PD parameters. This concept generalizes the idea behind receptor-based interaction as suggested by Ariëns[28]. The PD interaction is parameterized by INT (maximum fractional change of the victims PD parameter caused by the perpetrator), EC50$_{\text{INT}}$ (interaction potency), and $H_{\text{INT}}$ (interaction sigmoidicity).

For example, for two drugs A and B with a competitive-type interaction (EC50-level), the drug effects $E_A$ and $E_B$ are given by

$$E_A = \frac{\text{Emax}_A \times C_A^{H_A}}{\left(\text{EC50}_A \times \left(1 + \frac{\text{INT}_{AB} \times C_B^{H_{\text{INT,AB}}}}{\text{EC50}_{\text{INT,AB}}^{H_{\text{INT,AB}}} + C_B^{H_{\text{INT,AB}}}}\right)\right)^{H_A} + C_A^{H_A}}, \quad (1)$$

$$E_B = \frac{\text{Emax}_B \times C_B^{H_B}}{\left(\text{EC50}_B \times \left(1 + \frac{\text{INT}_{BA} \times C_A^{H_{\text{INT,BA}}}}{\text{EC50}_{\text{INT,BA}}^{H_{\text{INT,BA}}} + C_A^{H_{\text{INT,BA}}}}\right)\right)^{H_B} + C_B^{H_B}}, \quad (2)$$

INT$_{AB}$ represents the maximum fractional change of the EC50 of drug A (victim) caused by drug B (perpetrator), and vice versa for INT$_{BA}$. INT = 0 indicates no interaction, $-1 < \text{INT} < 0$ indicates a decrease of the EC50, and INT > 0 indicates an increase of the EC50. If both INT values are negative, synergy, and vice versa, antagonism is observed on the effect level. INT values of different polarities indicate an asymmetric interaction with concentration-dependent synergy and/or antagonism on the effect level. In addition, potentiation (inactive drug potentiates an active drug) or coalism (inactive drugs solely jointly active) can be modeled by the GPDI approach. Implementation of the GPDI model on EC50 leads to a competitive interaction behavior. An interaction of allosteric type is considered when the GPDI model is implemented on Emax. Note that the polarity of INT is opposite when implemented on Emax instead of EC50.

$E_A$ and $E_B$ are directly compatible with common additivity criteria to compute the combined effect including simple effect addition[29] ($E_{\text{comb}} = E_A + E_B$), Bliss Independence[21] ($E_{\text{comb}} = E_A + E_B - E_A \cdot E_B$) or highest single agent[2] ($E_{\text{comb}} = \max(E_A, E_B)$). The GPDI model is also compatible with Loewe Additivity, however solely for competitive-type interactions, as Loewe Additivity in its

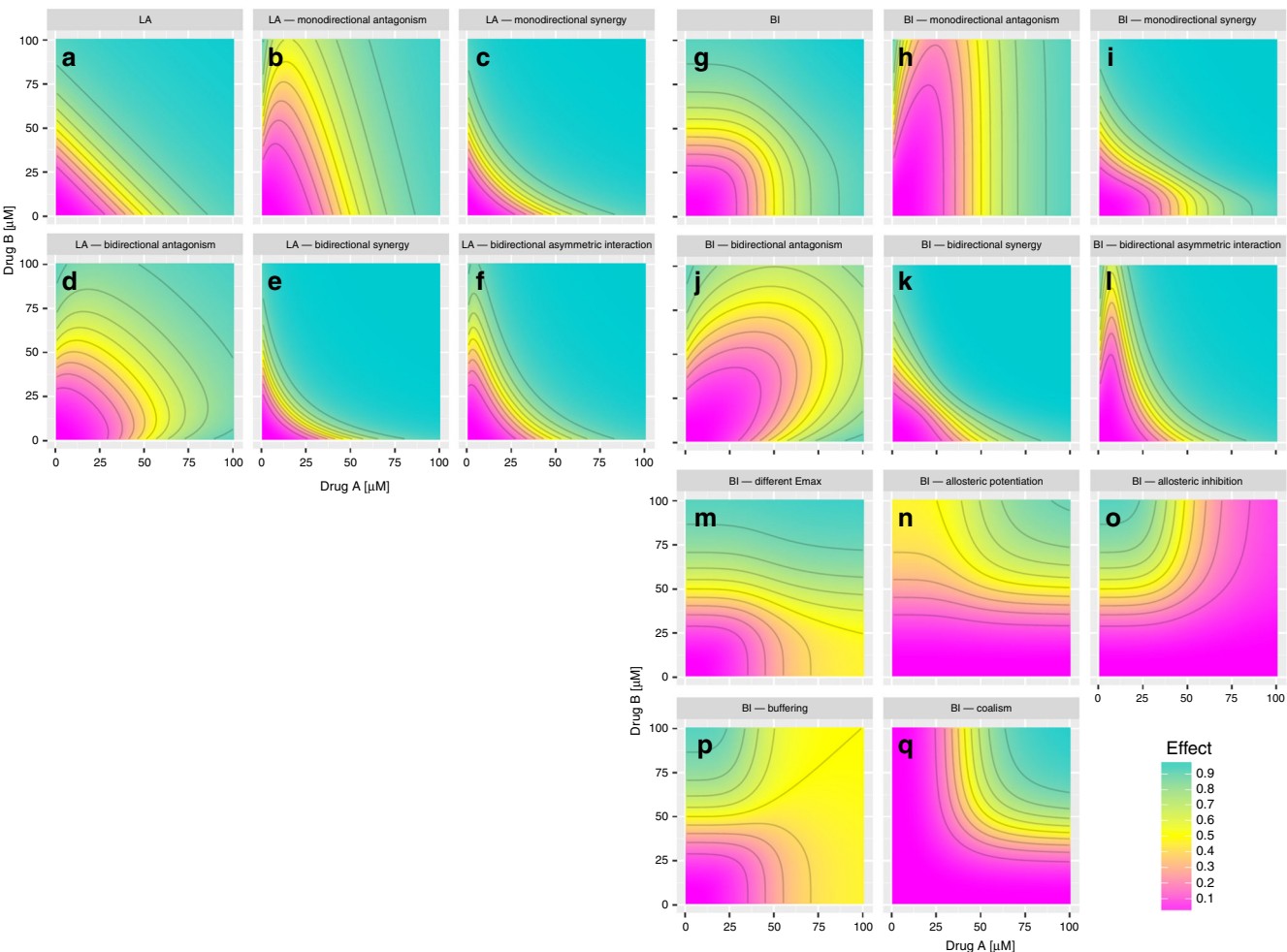

**Fig. 1** Simulated response surfaces using the GPDI model. Combined effects (color gradient) were simulated of drugs A and B for Loewe Additivity (LA, **a–f**) and Bliss Independence (BI, **g–q**) on EC50 level (**a–l**) or Emax level (**m–q**); Mono drug effects were parameterized with Emax$_A$ = Emax$_B$ = 1, EC50$_A$ = EC50$_B$ = 50 μM, $H_A$ = $H_B$ = 4 leading to null interaction surface for LA (**a**), and BI (**g**); monodirectional interaction surfaces (**b, c, h, i**), were obtained if A as perpetrator increased (**b, h**), or decreased (**c, i**), the EC50 of B; bidirectional interactions (**d–f, j–l**), were obtained if both drugs displayed the perpetrator role on each other with a joint increase of the EC50 leading to antagonism (**d, j**), or decrease of the EC50 leading to synergy (**e, k**), bidirectional asymmetric interactions (**b, d**), were obtained if A decreased EC50$_B$ and B increased EC50$_A$ leading to concentration-dependent synergy or antagonism; the GPDI model approach within Bliss Independence is compatible with different Emax values for both drugs (**m**) and can describe effects of itself inactive allosteric modulators (**n, o**), buffering (**p**), and coalism of two inactive drugs (**q**). The parameters of the GPDI model to generate scenarios are presented in Supplementary Table 1

original definition[20] requires a mutual maximum effect to exist:

$$1 = \frac{C_A}{EC50_A \times \left(1 + \frac{INT_{AB} \times C_B^{H_{INT,AB}}}{EC50_{INT,AB}^{H_{INT,AB}} + C_B^{H_{INT,AB}}}\right) \times \left(\frac{E_{comb}}{Emax_A - E_{comb}}\right)^{1/H_A}}$$
$$+ \frac{C_B}{EC50_B \times \left(1 + \frac{INT_{BA} \times C_A^{H_{INT,BA}}}{EC50_{INT,BA}^{H_{INT,BA}} + C_A^{H_{INT,BA}}}\right) \times \left(\frac{E_{comb}}{Emax_B - E_{comb}}\right)^{1/H_B}}, \quad (3)$$

Note that Loewe Additivity cannot be solved explicitly for $E_{comb}$ when EC50$_A$ ≠ EC50$_B$ and/or $H_A$ ≠ $H_B$, but implicitly by root finding methods, as outlined in the Online methods.

Signature plots of the GPDI model using Bliss Independence and Loewe Additivity are presented in Fig. 1 displaying the flexibility of the GPDI approach as it can describe very different types of PD interactions. The GPDI approach enables to describe monodirectional interactions with a defined perpetrator drug (Fig. 1b, c, h, i, n–p), bidirectional interactions with both drugs being perpetrator on each other (Fig. 1d, e, f, j–l, q), interactions with itself inactive drugs leading to allosteric interactions (Fig. 1n,

o), buffering (Fig. 1p) or coalism (Fig. 1q). We also confirmed the structural identifiability of the GPDI model in a conventional 8 by 8 checkerboard design with log2-diluitions typically done in, e.g., anti-infective PD interaction screening. Anticipated median relative standard errors (10–90th percentile) were 8.2% (2.8–29.8%) for INT and 20.6% (9.6–53.4%) for EC50$_{INT}$ (Supplementary Fig. 1). A log2 dilution design is preferable over a linear design. While still being identifiable, the anticipated median relative standard errors were 30.0% (11.1–268%) for INT and 52.4% (23.2–339%) for EC50$_{INT}$ in the linear design.

GPDI models for more than two interacting drugs follow the same principle of quantifying the interaction on the level of the PD parameter and solely require one additional GPDI term per added drug to quantify a bidirectional interaction. If a third interaction partner alters the interaction between two drugs, as potentially observable in triple combinations, e.g., emergent synergies[17], the interaction becomes tri-directional, and another interaction level (modulation) is added, i.e., a GPDI term on the INT parameter. Details on formulating the GPDI approach for more than two drugs are provided in the methods section.

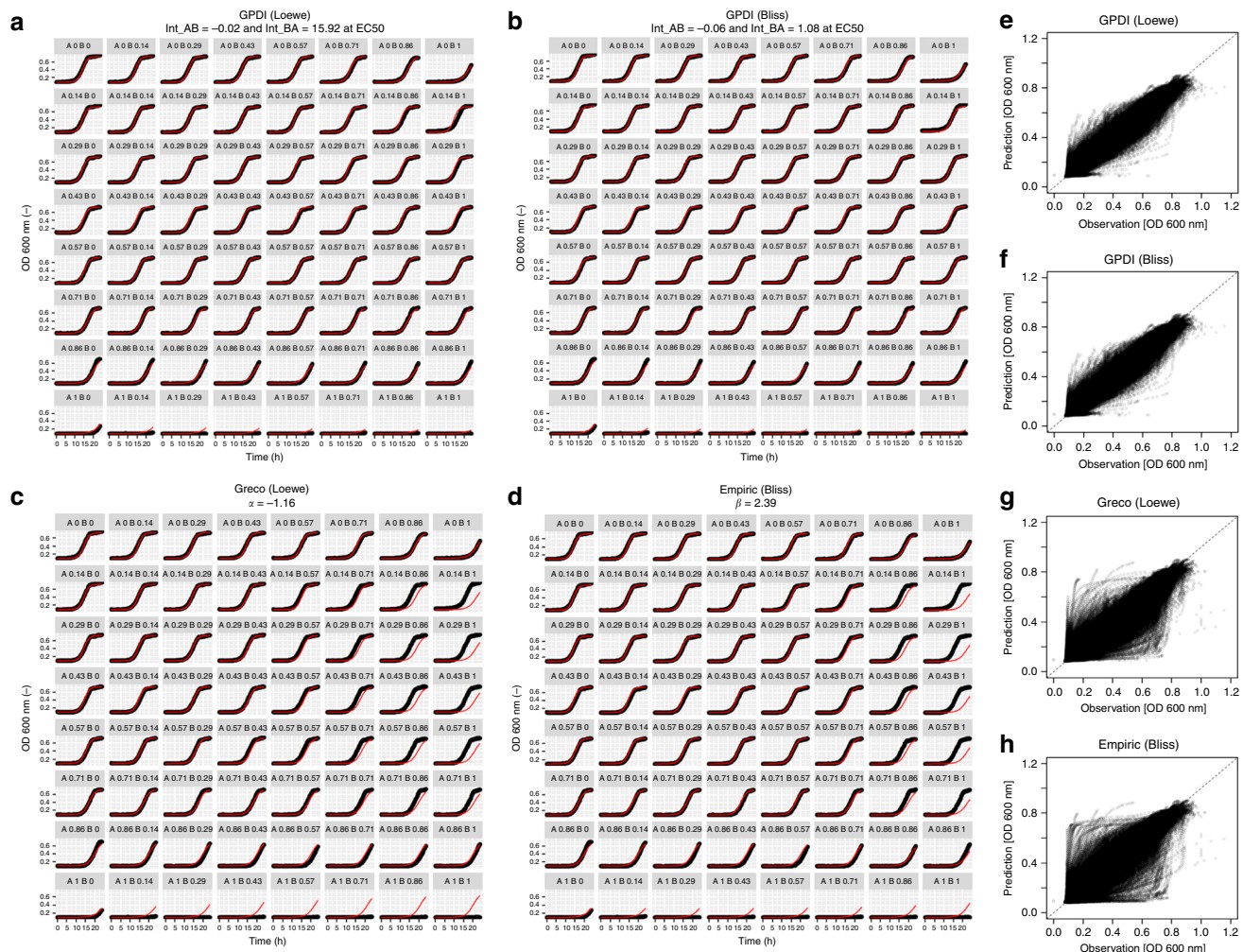

**Fig. 2** Individual model fit examples. The estimated parameters of the GPDI model were used for simulation (red) and compared to observed time-courses (black) exemplified in **a**–**d** for 1 out of the analyzed 200 combination scenarios; the combination of Bromopyruvate (Bro, drug A) and Staurosporine (Sta, drug B) (**a**–**d**), at concentrations AxBx (relative minimal inhibitory concentration[23]); The Loewe Additivity (**a**), and Bliss Independence-based (**b**), GPDI models described the 64 experiments of this combination scenario very well, the monodirectional nature of this interaction was indicated by the INT parameter values (fractional change of victim EC50) at the perpetrator EC50 that identified Bro as sole perpetrator drug in this combination; the empiric Loewe Additivity-based Greco (**c**), and empiric Bliss Independence (**d**), model solely quantified antagonism, but did not describe the data well, with prevalent with over- and underprediction; Predicted vs. observed fungal load for all 1.23 million observations from all 200 combination scenarios indicated superior predictive performance of the GPDI models (**e**, **f**), compared to the Greco and empiric Bliss model (**g**, **h**)

In situations with less rich data, the full four-parameter GPDI model can be reduced to adapt to these situations: a joint interaction term, i.e., $INT = INT_{AB} = INT_{BA}$ and/or interaction potency set to the drugs potency, i.e., $EC50_{INT} = EC50$, lead to a reduced one-parameter GPDI interaction model. It should be noted that the reduced GPDI models retain the interpretability of the interaction parameter. The GPDI model is applicable to other single drug exposure response models, e.g., slope or power effect model. Details for reduced models and other single drug exposure response models are provided in the Methods.

**Application of GPDI and comparison to conventional methods.** To evaluate the GPDI approach against conventional approaches, we used a pre-clinical study by Cokol et al.[23], who recorded the 24 h time-courses of fungal growth under perturbation of 200 antifungal and non-antifungal drug combinations. We fitted the Loewe Additivity- (Eq. (3)) and Bliss Independence-based GPDI models ($E_{comb} = E_A + E_B - E_A \cdot E_B$ with $E_A$ and $E_B$ given in Eq. (1) and Eq. (2)). Model building started with a reduced

GPDI model with a single interaction parameter INT and was extended to the full four-parameter GPDI model, which was significant ($\alpha = 0.05$, likelihood ratio test) in 152/200 scenarios (Loewe Additivity) or 167/200 (Bliss Independence). No interaction sigmoidicites ($H_{INT}$) were estimated. For the conventional models, we evaluated the isobole analysis[5,23] and the Greco model[24] as approaches based on Loewe Additivity, and an empiric Bliss Independence model[23]. The two different GPDI models provided an excellent performance to describe the observed time-courses of the (combined) effects, and were highly superior to the alternative models which did not describe the observed data well (Fig. 2, see Supplementary Data 1 for all evaluated 200 combination scenarios). This could also be objectified by assessing the respective distributions of the Akaike information criterion (AIC) (lower value indicates better model fit): AIC was in median (10–90th percentile) −29901.71 (−36805.3; −23776.7) for the Loewe Additivity-based GPDI model and −30377.43 (−37751.6; −24823.2) for Bliss Independence-based GPDI model. Hence, the Loewe Additivity- and Bliss Independence-based GPDI models described the

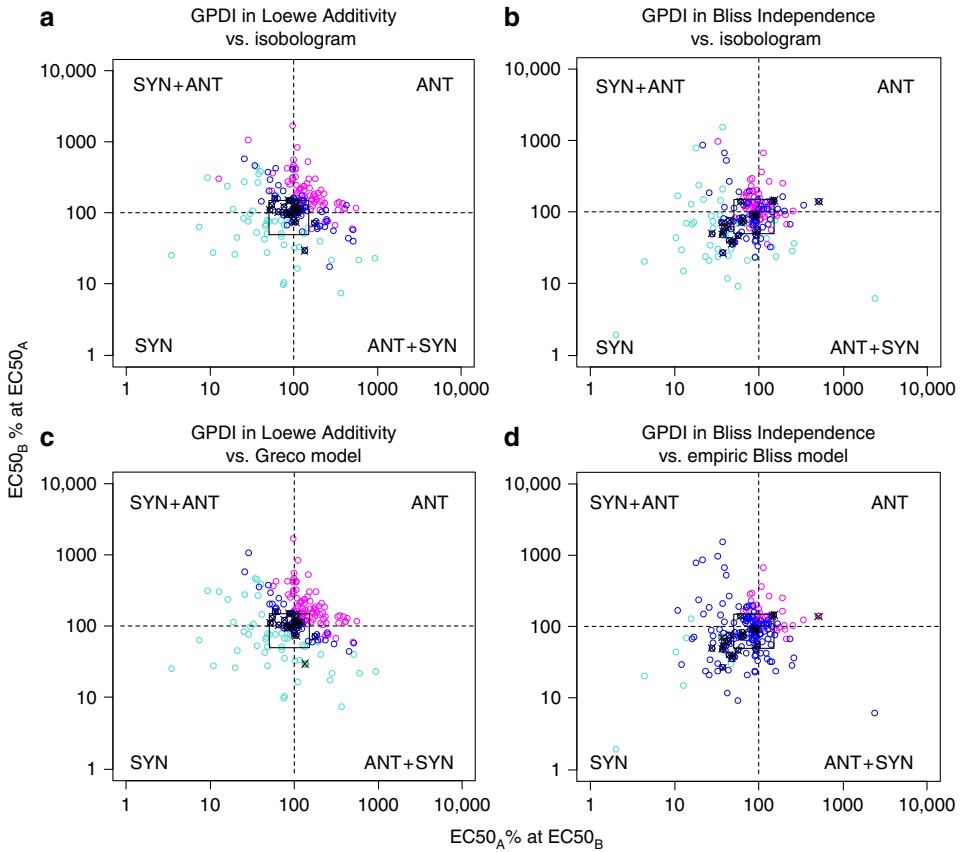

**Fig. 3** Distribution of the interaction parameters of the GPDI model. For simple interpretation, GPDI parameters are presented as %-change of the EC50 of the victim drug caused by the perpetrator drug at EC50 (points); dashed lines indicate additivity for Loewe Additivity (**a**, **c**), and Bliss Independence (**b**, **d**); the additivity margin (square) was determined from sham combinations (crossed points) with Loewe Additivity; the sectors indicate bidirectional synergy (SYN), bidirectional antagonism (ANT) or the two possible asymmetric interactions (SYN + ANT); color coding indicates result from corresponding conventional interaction analysis, i.e., isobologram (**a**, **b**), Greco model (**c**), and empiric Bliss model (**d**), and their determined interaction, i.e., synergy (turquoise), additivity (blue) or antagonism (magenta)

experimental data similarly well and were superior over the Greco model and the empiric Bliss Independence model which provided median AIC of −26989.4 (−35033.9; −17586.3) and −26152.5 (−33856.2; −18616.7), respectively. The modeled vs. observed time-courses of the PD interactions of all 200 scenarios for the GPDI and the conventional methods are provided in Supplementary Data 1. It should be noted that the conventional models solely have a single interaction parameter while the GPDI approach has up to four, but the high rate of statistical significance of the four parameter GPDI model in conjunction with the lower AIC values indicate that the additional parameters are highly supported. We did not evaluate allosteric-type interactions in the Cokol data set, as the "true" maximum antifungal effect may be fungicidal and hence beyond the turbidity threshold indicative merely for fungistasis.

For both Loewe Additivity and Bliss Independence-based GPDI models, in 199/200 scenarios, at least one statistically significant INT parameter was indicated by the likelihood ratio test. The full GPDI model was identified in ≥152/200 scenarios. As pure statistical significance can be misleading in rich-scale data, and to account for experimental variability in the data set, we used 25 sham combination experiments (drug A = drug B) in the Cokol data set to define an additivity margin for INT to conclude if significant antagonism or synergy was quantified: For the Loewe Additivity-based GPDI model, INT at the EC50 of the perpetrator was in median 0.025 (10–90th percentile: −0.47 to 0.5). Hence, the additivity margin was defined to lay within a

57–150% change of the victim EC50 at the EC50 of the perpetrator drug. For simplicity, we set the additivity margin to −0.5 to 0.5 for our analysis (cf. square in Fig. 3a–d). We used the same additivity margin in the Bliss Independence-based GPDI analysis due to the identical nature of the INT parameter, since there was a tendency to synergy of the INT values in the sham combinations with Bliss Independence (median: −0.378, 10–90th percentile: −0.639, 0.452). Self-synergy is an expected property of the Bliss Independence criterion as a drug's action cannot be independent from itself[10]. For the conventional methods, the interaction parameter $\alpha$ of the Greco model for the sham combinations was in median 0.23 and the 10–90th percentile (−0.45, 0.82) was used as additivity margin. For the empiric Bliss model, the interaction parameter $\beta$ of the sham combinations and hence the additivity margin ranged from −14,836 to 1.49 (10–90th percentile) and was −42.2 in median, hence as expected also displayed a tendency to self-synergy.

In the Cokol data set, 175 of the 200 scenarios comprised "true" combination experiments. For the Loewe Additivity-based GPDI analysis, the following scenarios were quantified: Additive interactions, i.e., both INT parameters within the additivity margin, were observed in 38/175 of the scenarios.

Two types of symmetric interactions were observed in 19/175 scenarios: (i) bidirectional synergy, where both INT parameters indicated synergy (< −0.5), i.e., both drugs were perpetrator and victim at the same time reducing each the others' EC50 values were found in 6/175 scenarios, and (ii) bidirectional

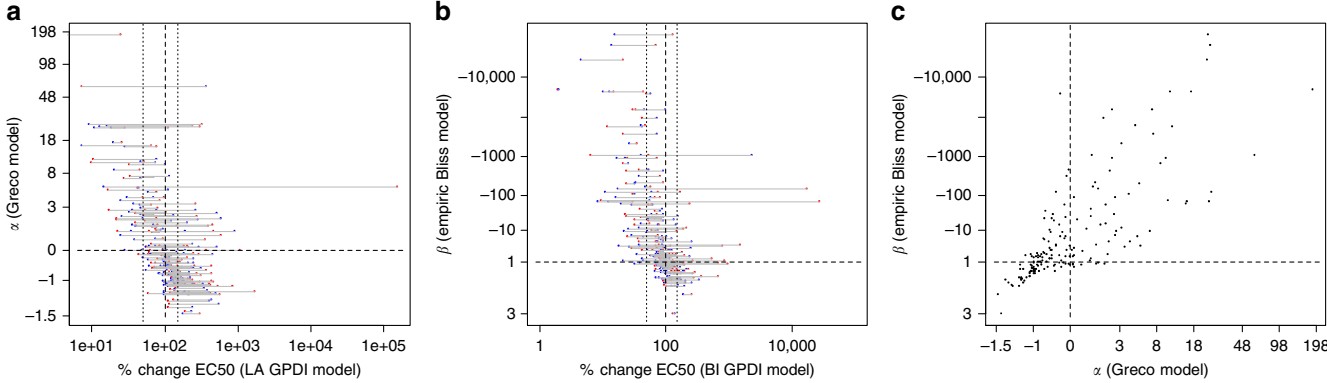

**Fig. 4** Comparison of the GPDI parameters to estimates of conventional methods. Scatter plots of the parameter $\alpha$ of the Greco model vs. the Loewe Additivity-based GPDI model parameters $INT_{AB}$, expressed as % change of the EC50 of drug A (red) and $INT_{BA}$, expressed as % change of the EC50 of drug B (blue) (**a**), the parameter $\beta$ of the empiric Bliss Independence model vs. the Bliss Independence-based GPDI model parameters (**b**), and $\alpha$ vs. $\beta$ (**c**); correlation of the parameters was ($r^2$) 0.26 (**a**), 0.24 (**b**), and 0.59 (**c**)

antagonism: Both INT parameters indicated antagonism (>0.5), i.e., both drugs are perpetrator and victim at the same time increasing each the others' EC50 values were found in 13/175 scenarios.

Three types of asymmetric interactions were observed in 118/175 scenarios: (i) monodirectional synergy, where one INT parameter indicated synergy (< −0.5), i.e., there was one perpetrator reducing the EC50 of the victim, while the other INT parameter was within the additivity margin, was observed in 20/175 scenarios, (ii) monodirectional antagonism, where one INT parameter indicated antagonism (>0.5), i.e., there was one perpetrator increasing the EC50 of the victim, while the other parameter was within the additivity margin, was observed in 76/175 scenarios, (iii) bidirectional asymmetric interactions, where both INT parameters were outside the additivity margin, but of opposite polarity, i.e., both drugs were perpetrator and victim at the same time increasing or decreasing each other's EC50 values leading to concentration-dependent antagonism or synergy, was observed in 22/175 scenarios.

While the overall conclusions on antagonism and synergy were in general agreement with the Loewe-based original isobole analysis of Cokol et al.[23] (Fig. 3a) and the result of the Greco model (Fig. 3c), it is evident that the direction of the interaction, i.e., the perpetrator–victim properties and the bidirectional asymmetric type interaction (cf. Fig. 1f) could not be classified with the conventional methods. 6 of these bidirectional asymmetric interaction were (wrongly) classified as additive, 14 as synergistic and 2 as antagonistic. Agreement between the Greco model and the isobole analysis was high and the approaches resulted in the same classification in 82% of the scenarios.

For the Bliss Independence-based GPDI analysis, a substantially higher synergy rate was found (monodirectional synergy: 30/175, bidirectional synergy: 14/175). Also, a higher rate of additive interactions was observed (79/175), 31/175 displayed monodirectional antagonism and only 2/175 interactions displayed bidirectional antagonism. 19/175 scenarios displayed bidirectional asymmetric interactions. The tendency to estimate a higher synergy rate with Bliss Independence is also apparent in Supplementary Fig. 2 when the estimated interaction parameters of Loewe Additivity and Bliss Independence-based GPDI models were compared. The Bliss Independence-based GPDI model did not agree with the isobole analysis, due to the different underlying additivity criterion (Fig. 3b). It agreed with the empiric Bliss Independence model (Fig. 3d) only when INT terms were of same polarity, but the empiric Bliss model was much less sensitive to detect interactions as indicated by the large number of

misclassified additive interactions. The parameter values of the GPDI models as well as for the alternative models are visualized and compared in Fig. 4. A re-analysis of the Cokol data set[30] revealed 61 suppressive interaction, i.e., strong antagonistic interactions with responses below one of either or both single agents. While a screen for suppressive interactions on the effect level using the GPDI approach is similar to the approach chosen by Cokol[30] (comparison of the combined effects with the single drug effects), the GPDI approach can enhance suppressive interactions with model-based estimates of perpetrator–victim information guided by the INT parameter. 66% of manually derived directions (assuming that the less potent drug antagonizes the more potent drug in these suppressive interactions[30]) were in agreement with the GPDI analysis, while in the remaining cases different and/or additional directions of the PD interactions were quantified (Supplementary Table 2). This indicates that the assumption that the less potent drug is the causative agent of suppression on the effect level is often, but not always true.

**Perpetrators and victims in a drug interaction network.** In order not to bias ourselves towards a single additivity criterion, we exclusively joined the result of the Loewe-Additivity and Bliss-Independence-based GPDI models for the network analysis, i.e., included solely interactions that were significant under both Loewe-Additivity and Bliss-Independence, which is displayed in Fig. 5a. The unjoined networks are presented in Supplementary Fig. 3. In the joined data set, 90 interactions (edges of the network) were observed where 54 of the interactions were monodirectional with one combination partner being perpetrator and the other being victim drug. 36 interactions were bidirectional, in which both combination partners were perpetrator and victim drug at the same time, and 12 out of the bidirectional interactions displayed INT values of joint polarity. 10 of these 12 were bidirectional synergies, e.g., as observed with the drugs Fen and Cal. 2 of these 12 were bidirectional antagonistic interactions (Cyc-Rad) and 24 interactions were bidirectional asymmetric interactions, e.g., drug Ben was antagonistic on drug Tac, but Tac potentiated the effect of Ben and concentration-dependent antagonism or synergy is observed (Supplementary Fig. 4). Another example is the drug Ter, which is part of a larger number of synergies. It has been speculated that Ter mediates synergy through its cell-wall disrupting effect which in turn might enhance the uptake of other drugs[23]. When comparing the GPDI parameters of interactions where Ter was involved, diverse interactions were observed (Supplementary Table 3). Yet, in the majority of the scenarios (5/11), Ter decreased the EC50 or did

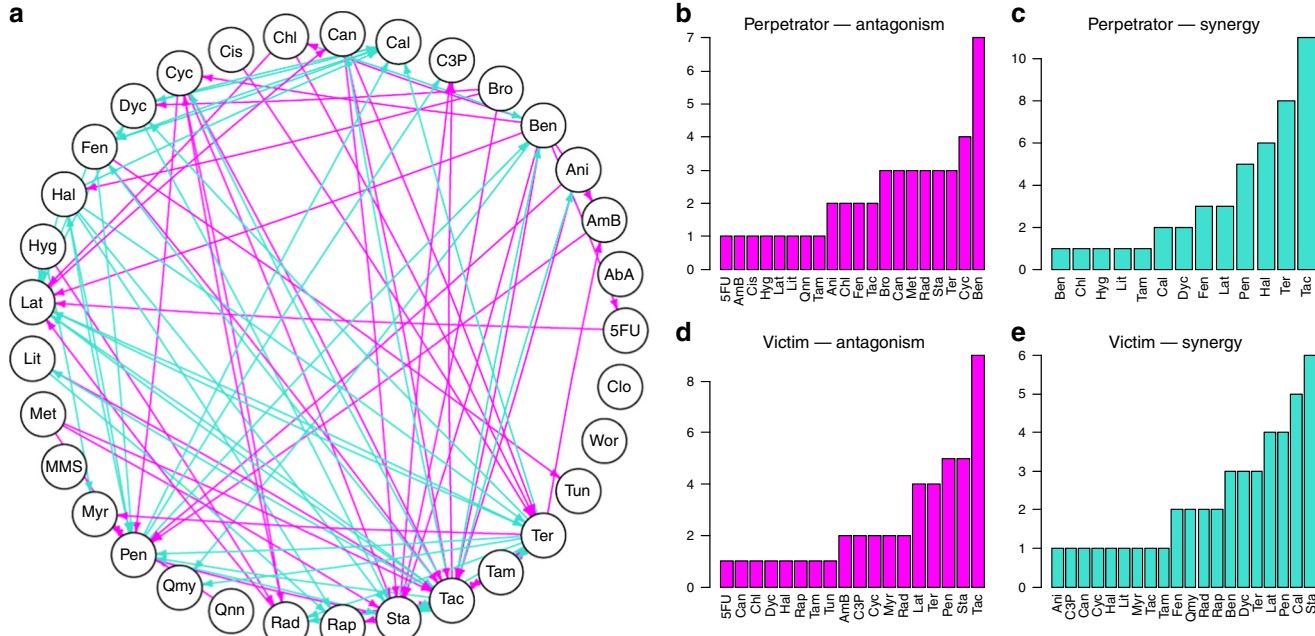

**Fig. 5** Perpetrator and victim behavior. Interaction network between the significant interactions from the exclusively joined Loewe Additivity and Bliss Independence-based GPDI analyses (**a**); arrows visualize direction of the PD interaction from the perpetrator drug to the victim drug, i.e., decrease of the victims EC50 resulting in antagonism (magenta) or increase of the victims EC50 resulting in synergism (turquoise); note that interactions can be mono-directional or bi-directional of same polarity (joint antagonism or joint synergy) or opposite polarity leading to asymmetric bidirectional interactions; frequency of perpetrator and victim behavior in the network is presented in **b**, to **d**. Abbreviations of the compounds: Aureobasidin A (AbA), Amphotericin B (AmB), Anisomycin (Ani), Benomyl (Ben), Bromopyruvate (Bro), CCCP (C3P), Calyculin A (Cal), Cantharidin (Can), Chlorzoxazone (Chl), Cisplatin (Cis), Clo (Clozapine), Cycloheximide (Cyc), Dyclonine (Dyc), Fenpropimorph (Fen), Haloperidol (Hal), Hygromycin (Hyg), Latrunculin B (Lat), Lithium (Lit), Methotrexate (Met), Methyl methanesulfonate (MMS), Myriocin (Myr), Pentamidine (Pen), Quinine (Qnn), Quinomycin (QMY), Radicicol (Rad), Rapamycin (Rap), Staurosporine (Sta), Tacrolimus (Tac), Tamoxifen (Tam), Tunicamycin (Tun), and Wortmannin (Wor)

not significantly affect the combination partners (4/11), which might corroborate the proposed mechanism, but also indicates that further yet unknown processes are affected which also alter the EC50 of Ter. Through its effect in the ergosterol pathway, Ter can also mediate monodirectional antagonism, e.g., as perpetrator on AmB, for which the INT value of 3.32 suggested an increase in EC50 to 432% ((1 + 3.32)*100%) at the EC50 of Ter, while the EC50 of Ter was not significantly altered. AmB binds ergosterol in the cell wall; it has been proposed that AmB might lose its target when ergosterol synthesis is inhibited[31], which would be in agreement with the observed INT values and the identified per-petrator role of Ter from the GPDI analysis. Of the mono-directional interactions, drugs Bro, Cis, Hyg, Met and Qnn were sole perpetrators being all antagonistic on their combination partners, but never took the victim role. For Bro, there is evidence that the antagonistic interactions observed with Bro are mediated by its acidity unrelated to its precise mode of action[30]. This distinct pattern is also found in the GPDI parameters: Bro increased the EC50 of its combination partners in 9/10 cases as perpetrator, while it was never affected itself, i.e., Bro mediated solely monodirectional antagonistic interactions (Supplementary Table 4). This indicates that the effects mediated by Bro itself might not be affected by the combination partners. Drugs Qmy, Tun, Rap, Myr and C3P were sole victim drugs, but never took the perpetrator role. Drug Tac was the most prevalent perpetrator for potentiation in 11 interactions, but 5 of those were antag-onistically "counteracted" by the victim. Drug Tac was also the most prevalent victim for antagonism in 9 interactions. For a large number of these interactions in the Cokol data set, the underlying mechanism of interaction remains to be elucidated. Further research is required to elucidate the molecular interaction

mechanisms to which the GPDI model might contribute by adding quantitative measures of EC50 shifts and perpetrator–victim information in order to profile the behavior of the drugs in interaction networks (Fig. 5b–e).

## Discussion

Accurate characterization of PD interactions is of key importance for defining rational combination therapies. In this paper, we derive the novel GPDI approach for this purpose. When para-meterizing the GPDI model parameters from the experimental data, the GPDI approach is complex enough to provide "spot-on" description of the observed time-courses of PD interactions in a data set of 200 combination experiments, including concentration-dependent interactions, but thereby still providing intuitive interaction parameters as fractional changes of the PD parameters and interaction potencies. For example, as illustrated in Fig. 2a, b, $INT_{AB}$ was determined to −0.02, i.e, the EC50 of drug A (Bro, bromopyruvate) was only marginally affected. Instead, $INT_{BA}$ was determined to 15.92, i.e., in combination the EC50 of drug B (Sta, staurosporine) was increased to 1692% ((1 + 15.92)*100%) at the EC50 of Bro. It is obvious, that the parameter $\alpha$ and $\beta$ of the conventional models (Fig. 2c, d) cannot be quantitatively interpreted in a similar way.

As the inferences are drawn on the level of the model para-meters and from all available scenarios of the checkerboard at the same time, no re-scaling on the effect level[16–18] is required, which is yet a helpful technique to score and detect interactions in individual scenarios. The GPDI approach revealed that the majority of up to 67% of the interactions in this data set were of asymmetric nature where distinct perpetrator and victim infor-mation in the PD interactions was quantified, i.e.,

monodirectional synergism, monodirectional antagonism, and asymmetric bidirectional interactions where synergism or antagonism depends on the concentration ratio of the drugs. This could not at all be classified by the conventional approaches[5,10,23] (Fig. 3). Concentration dependence in the PD interactions has also been a challenge in a recent study, that labeled these as "inconclusive"[32]. Moreover, the effect of itself inactive drugs on other active drugs could not be evaluated with conventional methods[23], which is possible with the GPDI approach.

The GPDI approach elicits the roles of perpetrator and victim in PD interactions and hence provided directed interaction networks as seen in Fig. 5. Such a directed interaction network opens up pivotal opportunities for cluster analyses to detect similarities of the drug's roles in large-scale interaction networks[32,33]. A recent study also addressed this important aspect of considering a direction in PD interactions[34] employing a two-parameter model. Yet, the approach they used is less versatile than the GPDI approach, as (i) they do not quantify an interaction on the level of the PD parameters, (ii) no modulation can be considered (cf. Eq. 12), (iii) their interaction parameter is not as easily interpretable, (iv) their approach is implemented solely in Bliss Independence, and (v) there model is likely less flexible than the GPDI approach which can use from one to four GPDI model parameters depending on the richness of the data. Higher-order combinations with three or more interacting drugs are not part of this work, but have been successfully modeled by statistical models[15]. Emergent synergies, i.e., synergistic effect only being present in three drug combination, but not in either dual combination[35], have been discovered recently. The GPDI model also can account for such "emergent interactions" through the modulator-term (Eqs. (11) and (12)). Recently, the presented GPDI model approach was applied in four drugs combinations in tuberculosis in vitro[36] and in vivo studies[37], where such interactions were also present and quantified by the GPDI model.

The GPDI approach advances receptor-based competitive or allosteric interaction models to the next evolutionary step by adding estimable interaction parameters for fractional change of the PD parameter and interaction potency, thereby being in accordance with the criteria for an ideal PD interaction model[12]. Moreover, the GPDI approach within Bliss Independence also covers scenarios which cannot be quantitatively modeled by conventional methodology such as pure potentiation or intertism, i.e., when solely one drug is active and its effect is or is not altered by an itself inactive drug, or coalism, i.e., when neither drug is active alone, but a combined effect is observed.

There is a controversy on which additivity criterion to use in PD interaction studies. Loewe Additivity[10,20] assumes that two drugs act through a similar mode of action, leading to the conception of "dose substitution", i.e., a fraction of drug A can be replaced by drug B leading to the same effect. Bliss Independence[10,21], however, assumes that drugs act independently through distinct mechanisms. The GPDI approach solves this controversy by unifying the interpretation of deviation from all these additivity criteria in a model-based framework. Our analysis of 200 combination experiments displayed that using Loewe Additivity or Bliss Independence can lead to different conclusions regarding synergy or antagonism on the observed interaction, yet by providing similar descriptive performance. With the GPDI approach, the analyst can decide on the most suitable interaction criterion for the present data, or as in the present work exclusively join the result of the modeling activities to select "strong" interactions that stick out under both additivity criteria. Yet, it should be noted that both the Loewe- and Bliss-based GPDI models described the observed data equally well. Hence, both would be suitable for performance of computer simulations, e.g., for therapeutic profiling of the PD interactions[38] from pre-clinical to

clinical phase or for clinical trial simulations, irrespective of the "true" underlying system. We also demonstrated that the investigated conventional approaches were not suitable to describe the data well and hence may not provide unbiased simulations if these models were used in translational predictions or combined dose finding.

The GPDI approach considers that PD might not be as simple as only synergism or antagonism, but the fact that PD interactions are a function of concentration and might be multi-dimensional, i.e., that the interaction changes in the concentration space. Moreover, the GPDI approach comes with further numerous advantages over existing approaches, which comprise quantitatively interpretable interaction point estimates across Loewe Additivity and Bliss Independence, no requirement of prior knowledge on the precise mode of (inter-)action, flexibility to adapt to multi-drug combination data of various complexity, compatibility with established additivity criteria, provision of insight into perpetrators and victims in PD interaction networks, and the possibility to describe time-courses of the interaction. Future studies should also evaluate the utility of the GPDI approach at other EC levels or the stationary concentration[39] as potency markers. Utilization of the GPDI approach in drug development opens up new perspectives for modeling, interpretation, quantitative decision marking and hypothesis generation in development of novel combination therapies.

## Methods

**The general pharmacodynamic interaction model.** We used the sigmoidal maximum effect model[25,26] for quantifying single drug effects. This model describes the drug effect $E$ as function of its concentration $C$ and the model parameters Emax (maximal drug effect), EC50 (drug concentration stimulating 50% of Emax, i.e., drug potency) and $H$ (Hill factor for sigmoidicity):

$$E(C) = \frac{\text{Emax} \times C^H}{\text{EC50}^H + C^H}. \tag{4}$$

The GPDI approach evolved from receptor-based models such as the competitive inhibition model, as proposed by Ariëns[28]

$$E_A(C_A, C_I) = \frac{\text{Emax}_A \times C_A}{\left(K_d \times \left(1 + \frac{C_I}{K_I}\right)\right) + C_A}, \tag{5}$$

expressing the induced effect $E_A$ of drug A at a concentration $C_A$ with a drug–receptor dissociation constant $K_d$ and an inhibitor I at a concentration $C_I$ and the inhibitor-receptor dissociation constant $K_I$. The GPDI model generalized the interaction term $(1 + C_I/K_I)$ using a maximum effect model with estimable interaction parameters (the "GPDI term") which allows both positive and negative shift of the PD parameter. The GPDI term is a function of the perpetrator drug concentration, which perturbs a PD parameter $\theta$ of the victim drug. The GPDI model is parameterized by INT representing the maximum possible fractional change of the PD parameter $\theta$, EC50$_{\text{INT}}$ which quantifies the potency of the perpetrator and $H_{\text{INT}}$, the sigmoidicity of the interaction effect.

$$\theta \times \left(1 + \frac{\text{INT} \times C^{H_{\text{INT}}}}{\text{EC50}_{\text{INT}}^{H_{\text{INT}}} + C^{H_{\text{INT}}}}\right). \tag{6}$$

The GPDI model considers that two drugs can potentially be both perpetrator and victim drug at the same time, hence quantifies bidirectional interactions. For two interaction partners with an interaction on the PD parameter EC50, the GPDI model is given by:

$$E_A = \frac{\text{Emax}_A \times C_A^{H_A}}{\left(\text{EC50}_A \times \left(1 + \frac{\text{INT}_{AB} \times C_B^{H_{\text{INT},AB}}}{\text{EC50}_{\text{INT},AB}^{H_{\text{INT},AB}} + C_B^{H_{\text{INT},AB}}}\right)\right)^{H_A} + C_A^{H_A}}, \tag{7}$$

$$E_B = \frac{\text{Emax}_B \times C_B^{H_B}}{\left(\text{EC50}_B \times \left(1 + \frac{\text{INT}_{BA} \times C_A^{H_{\text{INT},BA}}}{\text{EC50}_{\text{INT},BA}^{H_{\text{INT},BA}} + C_A^{H_{\text{INT},BA}}}\right)\right)^{H_B} + C_B^{H_B}}, \tag{8}$$

INT can take values between $-1$ and $\infty$ and guides the direction of the PD interaction: zero indicates no interaction, a negative value between $-1$ and 0 indicates potentiation and a positive value between 0 and $\infty$ indicates inhibitory behavior of the perpetrator. If both INT values are between $-1$ and 0, synergy is observed. Also potentiation (one drug active) or coalism (inactive drugs solely jointly active) can be modeled by the GPDI approach. If both INT values are

positive, antagonism is observed. Implementation of the GPDI model on EC50 leads to a competitive interaction behavior. An interaction of allosteric type is considered when the GPDI model is implemented on Emax. Note that the polarity of INT is opposite when implemented on Emax instead of EC50.

The GPDI model is also applicable when the drug effect is described by a linear slope model ($E =$ SLOPE$\cdot C$) or power model ($E =$ SLOPE$\cdot C^H$), which are identified if the highest studied concentration is well below the EC50. As SLOPE represents Emax/EC50, the GPDI term is both valid in numerator or denominator. Depending on this choice, the INT value will have the same interpretation as fractional change of EC50 (denominator) or Emax (numerator) for the slope model. For the power model, the implementation in the denominator will only approximate the fractional change of EC50, as in the power model the slope represents Emax/EC50$^H$.

Given that the GPDI model quantifies PD interactions at the level of the PD parameters, it is compatible with commonly used additivity criteria, such as Loewe Additivity[20], Bliss Independence[21] or simple effect addition[2]. The equations for the GPDI terms in the different additivity criteria are provided below in the application study section.

**The GPDI model for multiple interaction partners.** The GPDI model can be extended to quantify drug interactions between drugs A to $n$ by additional GPDI model terms for each additional drug on each PD parameter $\theta_{A,...,m}$ (interaction sigmoidicities $H_{INT}$ not displayed to ease readability):

$$\theta_{A,...,m} \times \prod_{i=1}^{n}\left(1 + \frac{INT_{A,n} \times C_n}{EC50_{INT,A,n} + C_n}\right). \quad (9)$$

Hence, for three drugs A, B and C, the bi-directional GPDI model for $E_A$ is formulated as follows:

$$E_A = \frac{Emax_A \times C_A^{H_A}}{\left(EC50_A \times \left(1 + \frac{INT_{AB} \times C_B}{EC50_{INT,AB} + C_B}\right) \times \left(1 + \frac{INT_{AC} \times C_C}{EC50_{INT,AC} + C_C}\right)\right)^{H_A} + C_A^{H_A}}. \quad (10)$$

If a drug modulates the interaction between two other drugs, another level of interaction (modulation) using the GPDI model can be implemented. If drug C modulates the interaction between A and B, the effect of drug A in presence of B and C can be formulated as follows:

$$E_A = \frac{Emax_A \times C_A^{H_A}}{\left(EC50_A \times \left(1 + \frac{INT_{AB} \times \left(1 + \frac{INT_{AB|C} \times C_C}{EC50_{INT,AB|C} + C_C}\right) \times C_B}{EC50_{INT,AB} + C_B}\right) \times \left(1 + \frac{INT_{AC} \times C_C}{EC50_{INT,AC} + C_C}\right)\right)^{H_A} + C_A^{H_A}}. \quad (11)$$

The quantitative interpretation of the modulation parameter $INT_{AB|C}$ is comparable to the interpretation of $INT_{AB}$: a value of zero indicates no modulation, a negative value between −1 and 0 indicates a positive modulation, and a value between 0 and ∞ indicates and negative modulation of the interaction between A and B.

One modulation level can be added for each additional drug, e.g., in our example an additional GPDI model term of a fourth drug D on $INT_{AB|C}$:

$$E_A = \frac{Emax_A \times C_A^{H_A}}{\left(EC50_A \times \left(1 + \frac{INT_{AB} \times \left(1 + \frac{INT_{AB|C} \times \left(1 + \frac{INT_{(AB|C)|D} \times C_D}{EC50_{INT,(AB|C)|D} + C_D}\right) \times C_C}{EC50_{INT,AB|C} + C_C}\right) \times C_B}{EC50_{INT,AB} + C_B}\right) \times \cdots\right)^{H_A} + C_A^{H_A}}. \quad (12)$$

**Parameterization of the GPDI model from experimental data.** For estimation of the GPDI model parameters from PD drug interaction studies, we propose the following stepwise regression approach using, e.g., maximum likelihood estimation: in the first step, the combination data is excluded from the analysis and solely the single drug effects are characterized by Emax, EC50, and $H$.

In the second step, the single drug effect parameters are fixed and solely the duo combination data is analyzed. A reduced GPDI model with a single interaction parameter $INT = INT_{AB} = INT_{BA}$ and $EC50_{INT} = EC50$ is evaluated first. The evolution from the reduced to the full GPDI model with all interaction parameters is guided by goodness-of-fit evaluation and statistical significance. As reduced GPDI models are nested within the full GPDI model, in maximum likelihood estimation, the likelihood ratio test can guide if an additional GPDI parameter is significant; i.e., a 3.84 drop in the objective function value is required to support inclusion of an additional parameter at $\alpha = 0.05$. For duo combination data, the final mathematical model is obtained at this stage.

In the third step, if trio combinations are available, the parameters of the single drug effects and the GPDI parameters from step two are fixed and modulator terms (e.g., $INT_{AB|C}$) are tested on the trio data. Inclusion of modulator parameters is again guided by goodness-of-fit evaluation and statistical significance. Step three is repeated for each additional drug that to estimate further modulator terms. In each

step, a reduced GPDI model, as outlined in step two, is tested first and expanded in a stepwise manner.

In the final step, the parameters of the mathematical model identified in the previous step are all unfixed and estimated to determine parameter precision.

**Identifiability analysis of the GPDI model.** To assess the structural identifiability of the GPDI model parameters, we performed 1000 computer simulations in an 8 by 8 checkerboard design with concentration tiers ranging from 0 to 8 (arbitrary concentration unit) in base 2 logarithmic steps. Model parameters of the Bliss Independence-based GPDI model were randomly sampled from uniform distributions with Emax $\in$ {0.5, 1.0}, EC50 $\in$ {0.5, 2.0}, $H \in$ {1.0, 4.0}, INT $\in$ {−0.9, −0.5} ∪ {0.5, 20}, and EC50$_{INT} \in$ {0.1, 1.0}. The standard deviation on effect level was set to 0.03 resembling a typical variability observed as, e.g., in in vitro PD interaction studies with antibiotics[9]. A linear dilution scheme was also assessed where the EC50 values where placed in the 40–60% range of the highest concentration studied, while all other parameters were as described above. Anticipated relative standard errors of the GPDI model parameters were calculated from the expected Fisher information matrix FIM:

$$FIM = \frac{1}{\sigma^2}\left(J \cdot J^T\right), \quad (13)$$

where $\sigma^2$ is the residual variance and $J$ is the Jacobian, i.e., the matrix of first-order derivatives with respect to the GPDI model parameters. Anticipated standard errors were calculated from the square root of the diagonal elements of the inverse FIM and normalized by the parameter value to obtain relative standard errors.

**Comparison to conventional methods.** A published data set by Cokol et al.[23] was used to evaluate the performance of the GPDI model and conventional methods. The different models were fitted to the data set, the performance was evaluated by goodness-of-fit (modeled vs. observed time-courses of the (combined) drug effects), and the conclusions from the GPDI and conventional approaches were compared.

The Cokol data set comprises 200 drug–drug combination scenarios evaluating the growth of *Saccharomyces cerevisiae* under (joint) drug exposure of various drugs in an 8 by 8 checkerboard fashion. A longitudinal growth model[40] was formulated as ordinary differential equation system and used to describe fungal growth (measured by optical density (OD) at 600 nm, with OD=$S_1 + S_2$) over time with $k_{lag}$, the delay rate constant to quantify lag of onset of fungal growth, $k_{growth}$, the rate constant for fungal growth being limited by the carrying capacity $B_{max}$ and inhibited by the (joint) drug effect $E$:

$$\frac{dS_1}{dt} = -S_1 \times k_{lag}, \quad (14)$$

$$\frac{dS_2}{dt} = S_1 \times k_{lag} + S_2 \times k_{growth} \times \left(1 - \frac{S_2}{B_{max}}\right) \times (1 - E_{comb}). \quad (15)$$

The (joint) drug effect $E$ was computed by the GPDI model implemented into either Bliss Independence ($E_{comb} = E_A + E_B - E_A \cdot E_B$ with $E_A$ and $E_B$ given in Eq. (1) and Eq. (2)) or Loewe Additivity (Eq. (4)). PD interactions between the drugs in the data set were captured using the competitive-type GPDI approach with interaction parameters on the level of EC50 (Eq. (9)). Given that the read-out for the antifungal effect was optical density, no alteration of the maximum effect was visible as it occurred beyond the turbidity threshold; hence Emax was set to 1 (= full inhibition of growth) in the analysis of the Cokol data set and no allosteric interactions could be evaluated.

Conventional approaches derived from Loewe Additivity and Bliss Independence were also evaluated: The Greco model,[24]

$$1 = \frac{C_A}{EC50_A \times \left(\frac{E}{Emax_A - E}\right)^{1/H_A}} + \frac{C_B}{EC50_B \times \left(\frac{E}{Emax_B - E}\right)^{1/H_B}} + \frac{\propto \times C_A \times C_B}{EC50_A \times EC50_B \times \left(\frac{E}{Emax - E}\right)^{\left(\frac{1}{2H_A} \cdot \frac{1}{2H_B}\right)}}, \quad (16)$$

in which PD interactions were quantified by an empirical factor $\alpha$, in which $\alpha=0$ indicated Loewe Additivity, $\alpha < 0$ antagonism and $\alpha > 0$ synergy, and an empiric Bliss interaction model,[23]

$$E = E_A + E_B - \beta \times E_A \times E_B, \quad (17)$$

in which $\beta=1$ indicated Bliss Independence and $\beta > 1$ antagonism and $\beta < 1$ synergy were utilized, and $E_A$ and $E_B$ represent ordinary sigmoidal maximum effect models.

Moreover, we compared the GPDI results against the Loewe Additivity-based isobole analysis method performed in the original study[23]. In an isobole analysis, the shape of an isobole, i.e., a line of the same effect, is analyzed. A linear isobole indicates Loewe Additivity, an isobole bent towards the origin indicates synergy, and vice versa antagonism. This curvature was quantified by a score parameter $\gamma$ on the longest isophenotypic curve on the checkerboard that used the area under the

time-kill curve, normalized by the area of the growth curve as effect metric, with

$$\gamma = \log\left(\frac{x}{1-x}\right) - \log\left(\frac{y}{1-y}\right),\qquad(18)$$

where $x$ and $y$ are MIC-normalized drug concentration and $\gamma$ described the bend of the isobole. $\gamma = 0$ indicated linear isoboles, i.e., Loewe Additivity, and for $\gamma < 0$ or $\gamma > 0$, synergy or antagonism, respectively, was determined.

Parameter estimation and data processing was performed in "R" (version 3.2.4, R Foundation for Statistical Computing, Vienna, Austria). Differential equations were solved using the "deSolve" package[41,42] (version 1.13). To increase performance, differential equations were encoded in C, compiled as dynamically linked library (.dll) and linked to the "deSolve" interface. Model parameters were estimated with "optim" from "stats" (version 3.2.4) using maximum likelihood estimation. The GNU scientific library for C[43] was used to provide root finding functionality within the ODE system for the Loewe Additivity models.

The results of the GPDI models and conventional models were compared regarding their performance by comparing modeled vs. observed time-courses of the anti-fungal effects. The AIC was calculated as an objective measure of model fit. Moreover, the classification result of the GPDI models and conventional models, i.e., if an interaction was additive, antagonistic or synergistic was assessed. Therefore, the result of estimated interaction parameters $INT_{AB}$ and $INT_{BA}$ were plotted normalized to the EC50 of the perpetrator drug. Positive INT values indicated antagonism; negative INT values indicated synergy and INT parameters are of opposite polarity indicated asymmetric PD interactions where the apparent synergy or antagonism is a function of the concentration. The classification result of the conventional approaches was mapped to this $INT_{AB}$–$INT_{BA}$ plot by color coding.

**Network analysis**. For visualization of the interactions, we chose a network plot with vertices being the drugs and the directed edges representing the detected directed interactions in the data set. In order to focus on the important interactions, the INT values of the victim drug at EC50 of the perpetrator drug being outside the additivity margin from both the Loewe-Additivity- and Bliss-Independence-based GPDI analysis were exclusively joined, i.e., only the interaction sticking out from both additivity criteria were displayed. The "igraph"-package[44] (version 1.0.1) in "R" was used for creating this network. The drugs were profiled for perpetrator and victim behavior.

**Code availability**. The model code of the GPDI model in the "R" language is provided as Supplementary Software 1.

**Data availability**. The utilized data set in the present work has been previously published by Cokol et al.[23] and can be accessed in the supplement of their publication.

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

## Acknowledgements

The authors would like to acknowledge our colleagues Dr Elin Svensson and Dr Anders Kristofferson at the Dept. of Pharmaceutical Biosciences for helpful discussions on combination therapies and Rikard Nordgren for technical support. The research leading to these results has received funding from the Swedish Research Council (grant number 521-2011-3442) and the Innovative Medicines Initiative Joint Undertaking (www.imi.europe.eu) under grant agreement no. 115337, resources of which are composed of financial contribution from the European Union's Seventh Framework Programme (FP7/2007–2013) and EFPIA companies' in kind contribution.

## Author contributions

S.G.W. developed the GPDI approach and conducted the analysis. C.C., O.C. and U.S.H. S. contributed to the development of the GPDI approach and the analysis. All authors wrote and approved the manuscript.

## Additional information

**Competing interests:** The authors declare no competing financial interests.

