## [Peer Review File · Nature Communications]

Reviewers' comments:

Reviewer #1 (Remarks to the Author):

This is a very intriguing study set out to produce a model of drug interactions that can overcome a number of known limitations of previous work in this topic. I think the topic and the results certainly will be useful to the large group of people who work on drugs, and especially interesting to the smaller subset that study drug interactions. I find the topic compelling, and I am excited about this work, but ultimately I have several major (but addressable) issues the authors should consider:

1. The literature is rich in drug interactions and I found the treatment a bit spotty. A fuller introduction and background of this is needed. Indeed, there is not only previous work by Chou and Talalay, Berenbaum, but there is more recent work from Jaynes et al. 2012 *Statistics in Medicine*, Wood et al 2012 *PNAS*, Tekin et al 2015 *Interface*, Chevereau and Bollenbach 2015 *Mol Sys Bio*, Bollenbach 2015 *Current Opinion Microbiology*, Mitosh and Bollenbach 2014 *Env Microbiology Reports*, Baym, *Science* 2016.
2. There has been work on what people have called "re-scaling" and this seems to significantly affect results of drug interactions. See Segre et al 2005 *Nature Genetics* and Tekin et al. 2016. *Royal Society Interface*. To the best of my understanding of the models presented here, this model does not consider "re-scaling" which appears to alter significantly findings of interactions. Revision to include "re-scaling" would be crucial.
3. I find the explanation of 'perpetrator' and 'victim' incomplete at best, especially as I understand it, these are new terms being introduced by the authors? If not, there should be references given. I think a much better description of what is a perpetrator, what is a victim, and how do you define these and tease these apart is needed.
4. More explicit comparison to the recently published multiple antibiotic papers that have come out (Zimmer et al 2012 *PNAS* was discussed briefly but I think more information on the applicable differences – for example, it seems one of the advantages to your model is there are fewer parameters to fit?) but also a newly published one (Beppler et al *Interface* 2016) and a previous one (Wood et al *PNAS* 2012).

Reviewer #2 (Remarks to the Author):

The study by Wicha et al proposes a new drug interaction scoring model, which also allows the quantification of asymmetric interactions. The authors use their model on previously published experimental data and describe their findings. Asymmetric interactions such as drug suppression has been previously analyzed using Bliss Model (Cokol 2014). However, a more coherent framework is needed for this purpose, and as the authors point out Loewe method cannot address it. While I am enthusiastic about a framework to analyze asymmetric drug interactions, the manuscript at its present form is difficult to follow and is not suitable for a broad audience. In addition, I have a number of concerns.

- As far as I understand, the authors never predict an interaction. They parameterize an observed checkerboard assay using the GPDI method. This is not different than scoring an interaction with Loewe or Bliss Models. If so, then the authors should edit their statements about predictions.
- The concepts of perpetrator and victim drugs are central to the study. However, their definition is vague and their interpretation is not well explained.

- The four parameters for the GPDI model is not well described. Authors may want to write the GPDI values on the simulations in Figure 1.

Line 40: Authors claim other methods are not "conclusively interpretable", yet Fractional Inhibitory Concentration has a simple interpretation. Also, after reading the manuscript, I am unsure if GPDI is conclusively interpretable. Authors should either clarify the model parameter interpretations or tune down their claim of interpretability.

Line 49,50: Point vi is unclear. How can drug development process effect a drug interaction?

Line 51: Perpetrator and victim is not yet defined.

Line 57: Perpetrator and victim is not mentioned in Yeh 2006.

Line 82: competitive-type interaction not defined.

Line 124: How is the significance calculated?

Line 129: GPDI is a four-parameter model, while these models have one parameter. Is this comparison fair?

Line 161-163: Although their values are different, authors can compare them using rank based methods.

Line 172-175: It seems unfair that authors claim Cokol 2011 misclassified these interactions, since that study did not look for asymmetric interactions. A better comparison might be Cokol 2014, which reported suppressive interactions using the same data set as the authors use.

Line 194: Authors should define exclusive joining.

Line 219: I don't believe this is intuitive. Manuscript would benefit a lot from examples for intuition.

- How do the INT values and interaction scores from other methods compare? Authors should include this as scatter plots.

Line 250-252: Authors previously claimed that these methods give similar answers.

- The arrows in Figure 4a are too small. How are these arrows interpreted?

- Can the authors comment on why some drugs show perpetrator or victim behavior?

Point-by-point response to the revision of the manuscript:

On perpetrators and victims: A general pharmacodynamic interaction model identifies the protagonists in drug interaction studies.

Sebastian G. Wicha, Chunli Chen, Oskar Clewe and Ulrika S.H. Simonsson

Nature Communications, NCOMMS-16-26282-T

Line numbers of the reviewers refer to the original submission, line numbers of our responses and changes to the manuscript refer to the revised version of the manuscript incl. track-changes.

Reviewer #1 (Remarks to the Author):

Reviewer #1: This is a very intriguing study set out to produce a model of drug interactions that can overcome a number of known limitations of previous work in this topic. I think the topic and the results certainly will be useful to the large group of people who work on drugs, and especially interesting to the smaller subset that study drug interactions. I find the topic compelling, and I am excited about this work, but ultimately I have several major (but addressable) issues the authors should consider:

Thank you for this overall positive feedback on our work and the constructive criticism and feedback for improving our manuscript. We provide our responses and potential changes to the manuscript below.

Reviewer #1: 1. The literature is rich in drug interactions and I found the treatment a bit spotty. A fuller introduction and background of this is needed. Indeed, there is not only previous work by Chou and Talalay, Berenbaum, but there is more recent work from Jaynes et al. 2012 Statistics in Medicine, Wood et al 2012 PNAS, Tekin et al 2015 Interface, Chevereau and Bollenbach 2015 Mol Sys Bio, Bollenbach 2015 Current Opinion Microbiology, Mitosh and Bollenbach 2014 Env Microbiology Reports, Baym, Science 2016.

We added the mentioned approaches to the introduction where appropriate. We agree that the overview on the landscape of current approaches is now much more complete.

Changes to the manuscript:

I. 64-69:

Recent and current work is addressing these limitations e.g. through more efficient factorial designs¹, innovative mechanism-independent statistical models², re-scaling techniques to facilitate correct interaction scoring in two³ and three drug combinations^{4,5}. Moreover, the genetic impact on drug interactions is also increasingly studied⁶ to assess the genetic robustness of PD interactions.

Yet, all these efforts could be complemented by a

The objective of the present work was to define a general pharmacodynamic interaction (GPDI) model overcoming all these limitations (i-vii).

Reviewer #1: 2. There has been work on what people have called “re-scaling” and this seems to significantly affect results of drug interactions. See Segre et al 2005 Nature Genetics and Tekin et al. 2016. Royal Society Interface. To the best of my understanding of the models presented here, this model does not consider “re-scaling” which appears to alter significantly findings of interactions. Revision to include “re-scaling” would be crucial.

This is an important aspect. We thank the reviewer for pointing to this. We agree that rescaling is a helpful technique when interactions are directly classified from the level of drug effects or bacterial fitness measures of individual experiments. To relate our work to the re-scaling approach, re-scaling is defined in this example for Bliss Independence where an observed combined response E_{AB} is compared to the product of individual effects E_A and E_B ³:

$$\epsilon = E_{AB} - E_A * E_B$$

In rescaling, the term ϵ is then normalized by $\text{abs}(E_{AB} - E_A * E_B)$ and E_{AB} is set to either $\min(E_A, E_B)$ when $E_{AB} > E_A * E_B$ or to zero otherwise. Due to rescaling, antagonisms cluster at ~ -1 , no interactions at ~ 0 and synergies at ~ 1 when individual experiments are assessed.

Re-scaling is not required and would be even counterproductive in the context of the GPDI model: The GPDI model is a mathematical model that describes the entire interaction surface, i.e. evaluates all individual experiments of a drug interaction experiment at the same time incl. their time- and concentration-dependence. The GPDI model has demonstrated in this study that it can actually describe and quantify the differences, i.e. ϵ . Normalisation of these differences to -1, 0 and 1 not be helpful in this case.

Another difference to rescaling is that the inference on an interaction is drawn on level of the parameters of the GPDI model, i.e. shift of the victims EC50 (or Emax) by the perpetrator drug. Thereby, a more global and mechanistic conclusion on a pharmacodynamic interaction can be drawn than from individual assessment of re-scaled ϵ values. We still think the re-scaling approach is very valuable for assessing individual experiments though, yet it is not required in the context of the GPDI approach.

We feel that these aspects will be very important to future readers of the manuscript, so we added this aspect.

Changes to the manuscript:

I. 326-328:

As the inferences are drawn on the level of the model parameters and from all available scenarios of the checkerboard at the same time, no re-scaling on the effect level^{3,5} is required, which is yet a helpful technique to score and detect interactions in individual scenarios.

Reviewer #1: 3. I find the explanation of ‘perpetrator’ and ‘victim’ incomplete at best, especially as I understand it, these are new terms being introduced by the authors? If not, there should be references given. I think a much better description of what is a perpetrator, what is a victim, and how do you define these and tease these apart is needed.

Thank you. We agree that this is important and the definition was not clear enough if readers are not familiar with the terminology. We are not aware that perpetrator and victim terminology has been used in the context of PD interactions before. Yet, these terms are common in the context of

pharmacokinetic (PK) interactions. We added a definition and reference to the terminology in the PK context.

Changes to the manuscript:

I. 87-104:

Single drug effects were characterized with the sigmoidal maximal effect (E_{max}) model^{7,8}, parameterized by PD parameters E_{max} (maximal drug effect), $EC50$ (drug concentration stimulating 50% of E_{max} i.e. drug potency) and H (Hill factor for sigmoidicity). The concept of the GPDI approach is simple: We propose a PD interaction to be quantifiable as shift in E_{max} (allosteric type) or $EC50$ (competitive type), which provides an intuitive, mechanistically-motivated, quantitative, and statistically interpretable (point) estimate of a PD interaction. A central aspect to the GPDI model is its ability to define perpetrators and victims of a PD interaction: A perpetrator alters the PD parameter of the victim drug leading to a PD interaction, i.e. either synergy or antagonism. The interactions in the GPDI approach are bi-directionally quantified, i.e. the drug can take the role of perpetrator, victim, or even both at the same time. This definition of perpetrator and victim is to our best knowledge new in the context of PD interactions, yet similar to the use of such terms in the context of pharmacokinetic (PK) interactions, where e.g. drug elimination of a victim drug is reduced by a perpetrator drug⁹. To include PD interactions, we extended the sigmoidal E_{max} model of the victim drug by a perpetrator sigmoidal E_{max} term ('GPDI term') to capture the interaction effect on the level of the PD parameters. This concept generalizes the idea behind receptor-based interaction as suggested by Ariëns¹⁰. ~~The interactions in the GPDI approach are bi-directionally quantified, i.e. the drug can take the role of perpetrator, victim, or even both at the same time.~~

Reviewer #1: 4. More explicit comparison to the recently published multiple antibiotic papers that have come out (Zimmer et al 2012 PNAS was discussed briefly but I think more information on the applicable differences – for example, it seems one of the advantages to your model is there are fewer parameters to fit?) but also a newly published one (Beppler et al Interface 2016) and a previous one (Wood et al PNAS 2012).

We now provide further discussion on the model developed by Zimmer et al and particularly added the aspect of the number of parameters. It should be noted that the GPDI model per se does not require four parameters. As outlined in details in the methods section, it is also possible to use a 2 parameter GPDI model, e.g. by setting the $EC50$ of the interaction to the $EC50$ of the drug. The discussion of the work by Beppler et al and Wood et al was also expanded.

Changes to the manuscript:

I. 339-346:

~~We appreciate a~~ A very recent study also addressed this important aspect of considering a direction in PD interactions¹¹ employing a two-parameter model. Yet, the approach they used is less versatile than the GPDI approach, as (i) they do not quantify an interaction on the level of the PD parameters, (ii) no modulation can be considered (cf. **Fehler! Verweisquelle konnte nicht gefunden werden.**), (iii) their interaction parameter is not as easily interpretable ~~and~~, (iv) their approach is implemented solely in Bliss Independence, and (v)

their model is likely less flexible than the GPDI approach which can use from one to four GPDI model parameters depending on the richness of the data.

I. 346-352:

Higher-order combinations with three or more interacting drugs are not part of this work, but have been successfully modelled by statistical models². Emergent synergies, i.e. synergistic effect only being present in three drug combination, but not in either dual combination¹², have been discovered recently. The GPDI model also can account for such 'emergent interactions' through the modulator-term (Eq. 11 and 12). Recently, the presented GPDI model approach was applied in four drugs combinations in tuberculosis in vitro¹³ and in vivo studies¹⁴, where such interactions were also present and quantified by the GPDI model.

I.140-141:

If a third interaction partner alters the interaction between two drugs, as potentially observable in triple combinations, e.g. emergent synergies⁴, the interaction becomes tri-directional, and another interaction level (modulation) is added, i.e. a GPDI term on the *INT* parameter.

Reviewer #2 (Remarks to the Author):

Reviewer #2: The study by Wicha et al proposes a new drug interaction scoring model, which also allows the quantification of asymmetric interactions. The authors use their model on previously published experimental data and describe their findings. Asymmetric interactions such as drug suppression has been previously analyzed using Bliss Model (Cokol 2014). However, a more coherent framework is needed for this purpose, and as the authors point out Loewe method cannot address it. While I am enthusiastic about a framework to analyze asymmetric drug interactions, the manuscript at its present form is difficult to follow and is not suitable for a broad audience. In addition, I have a number of concerns.

Thank you for your positive feedback on the importance of creating a new framework to detect asymmetric interactions, which we presented in the submitted manuscript. In order to make the manuscript, i.e. the communication of our model and the derived results easier to follow and more suitable for a wide audience, we added a more detailed definition of perpetrators and victims (see your comment and our response below).

In addition, we also relate the perpetrator-victim properties of the drugs to observing symmetric or asymmetric interactions, which definitions were refined in this revision. We agree that this aspect of our work could be presented more clearly. To summarize, the following scenarios are quantifiable with the GPDI approach:

- (i) Additive interaction: Both INT parameters are within the additivity margin (in this study determined to -0.5 to 0.5 by sham combinations).
- (ii) Symmetric interactions:
 - a. Bidirectional synergy: Both INT parameters are indicating synergy, i.e. both drugs are perpetrator and victim at the same time reducing each their EC50 values.

- b. Bidirectional antagonism: Both INT parameters are indicating antagonism, i.e. both drugs are perpetrator and victim at the same time increasing each their EC50 values.
- (iii) Asymmetric interactions:
 - a. Monodirectional synergy: One INT parameter indicates synergy, i.e. there is one perpetrator reducing the EC50 of the victim, while the other parameter is within the additivity margin.
 - b. Monodirectional antagonism: One INT parameter indicates antagonism, i.e. there is one perpetrator increasing the EC50 of the victim, while the other parameter is within the additivity margin.
 - c. Bidirectional asymmetric interaction: Both INT parameters are outside the additivity margin, but of opposite polarity, i.e. both drugs are perpetrator and victim at the same time increasing or decreasing each other's EC50 values leading to concentration-dependent antagonism or synergy.

In the previously submitted version, we only assessed the polarity of the INT values for defining an asymmetric interaction. We feel that this updated, more detailed definition presented above provides a much more detailed insight into the possible interactions we quantified in the Cokol dataset using the GPDI model. We added these updated definitions along with the provision of more detailed results to the manuscript:

I.202-254:

For the Loewe additivity-based GPDI analysis, the following scenarios were quantified: Additive interactions, i.e. both INT parameters within the additivity margin, were observed in 38/175 of the scenarios.

Two types of symmetric interactions were observed in 19/175 scenarios:

(i) bidirectional synergy, where both INT parameters indicated synergy (< -0.5), i.e. both drugs were perpetrator and victim at the same time reducing each the others' EC50 values were found in 6/175 scenarios, and

(ii) bidirectional antagonism: Both INT parameters indicated antagonism (> 0.5), i.e. both drugs are perpetrator and victim at the same time increasing each the others' EC50 values were found in 13/175 scenarios.

Three types of asymmetric interactions were observed in 118/175 scenarios:

(i) monodirectional synergy, where one INT parameter indicated synergy (< -0.5), i.e. there was one perpetrator reducing the EC50 of the victim, while the other INT parameter was within the additivity margin, was observed in 20/175 scenarios,

(ii) monodirectional antagonism, where one INT parameter indicated antagonism (> 0.5), i.e. there was one perpetrator increasing the EC50 of the victim, while the other parameter was within the additivity margin, was observed in 76/175 scenarios,

(iii) bidirectional asymmetric interactions, where both INT parameters were outside the additivity margin, but of opposite polarity, i.e. both drugs were perpetrator and victim at the same time increasing or decreasing each other's EC50 values leading to concentration-dependent antagonism or synergy, was observed in 22/175 scenarios.

For the Loewe Additivity based GPDI analysis, 38/175 combinations interacted additively, 20/175 combinations displayed joint decrease of EC50s, i.e. bidirectional synergism, 51/175 combinations displayed joint increase of EC50s, i.e. bidirectional antagonism, which While the overall conclusions on antagonism and synergy was in general agreement with the Loewe-based original isobole analysis of Cokol et al¹⁵ (Figure 3a) and the result of the Greco model (Figure 3c), it is evident that the direction of the interaction, i.e. the perpetrator-victim properties and the bidirectional asymmetric type interaction (cf. Figure 1f) could not be classified with the conventional methods. 6 of these bidirectional asymmetric interaction were (wrongly) classified as additive, 14 as synergistic and 2 as antagonistic. Yet, 66/175 combinations displayed

~~asymmetric interactions, i.e. INT terms of opposite polarity (c.f. Figure 1f). In these, it will depend on the concentration ratio of A and B if antagonism or synergy is apparent. This interaction type was not at all captured by the conventional methods: 26/66 of these asymmetric interactions were wrongly classified as additive, 20/66 as synergistic and 20/66 as antagonistic. Hence, 66/175, i.e. 38% of the interactions in the Cokol dataset were wrongly classified by using conventional isobole interaction assessment. Similar misspecification was observed for the Greco model (Figure 3e).~~ Agreement between the Greco model and the isobole analysis was high and the approaches resulted in the same classification in 82% of the scenarios.

For the Bliss Independence-based GPDI analysis, a substantially higher synergy rate was found (monodirectional synergy: 3630/175, bidirectional synergy: 14/175). Also, a higher rate of additive interactions was observed (79/175) and 31/175 displayed monodirectional antagonism and only 102/175 interactions displayed joined bidirectional antagonism. 5019/175 scenarios displayed bidirectional asymmetric interactions. The tendency to estimate a higher synergy rate with Bliss Independence is also apparent in Figure S2 when the estimated interaction parameters of Loewe Additivity and Bliss Independence-based GPDI models were compared. The Bliss Independence-based GPDI model did not agree with the isobole analysis, due to the different underlying additivity criterion (Figure 3b). It agreed with the empiric Bliss Independence model (Figure 3d) only when INT terms were of same polarity, but the empiric Bliss model was much less sensitive to detect interactions as indicated by the large number of misclassified additive interactions. ~~Most of the asymmetric interactions from the Bliss Independence-based GPDI model were in the empiric Bliss model classified as additive (34/50), 1/50 as synergistic and 15/50 as antagonistic.~~

Reviewer #2:- As far as I understand, the authors never predict an interaction. They parameterize an observed checkerboard assay using the GPDI method. This is not different than scoring an interaction with Loewe or Bliss Models. If so, then the authors should edit their statements about predictions.

This is an important aspect and we agree it could be made clearer. It is correct that the GPDI model does not predict PD interactions from other external experimental data. However, when parameterizing the GPDI model from the experimental data, the GPDI model is able to predict the observed PD interactions 'spot-on', incl. the time-course of the effects by using meaningful and interpretable interaction parameters. We changed the manuscript text accordingly to outline more clearly, what we mean by prediction.

I. 316-317:

When parameterizing the GPDI model parameters from the experimental data, The the GPDI approach is complex enough to provide 'spot-on' predictions of the observed time-courses of PD interactions in a dataset of 200 combination experiments, including concentration-dependent, i.e. asymmetric interactions, but thereby still providing intuitive interaction parameters as fractional changes of the PD parameters and interaction potencies.

Reviewer #2:- The concepts of perpetrator and victim drugs are central to the study. However, their definition is vague and their interpretation is not well explained.

Thank you. We agree that this is important and the definition was not clear enough if readers are not familiar with the terminology. We are not aware that perpetrator and victim terminology has been

used in the context of PD interactions before. Yet, these terms are common in the context of PK interactions. We added a definition and reference to the terminology in the PK context.

Changes to the manuscript:

I. 87-96:

Single drug effects were characterized with the sigmoidal maximal effect (E_{max}) model^{7,8}, parameterized by PD parameters E_{max} (maximal drug effect), $EC50$ (drug concentration stimulating 50% of E_{max} i.e. drug potency) and H (Hill factor for sigmoidicity). The concept of the GPDI approach is simple: We propose a PD interaction to be quantifiable as shift in E_{max} (allosteric type) or $EC50$ (competitive type), which provides an intuitive, mechanistically-motivated, quantitative, and statistically interpretable (point) estimate of a PD interaction. A central aspect to the GPDI model is its ability to define perpetrators and victims of a PD interaction: A perpetrator alters the PD parameter of the victim drug leading to a PD interaction, i.e. either synergy or antagonism. The interactions in the GPDI approach are bi-directionally quantified, i.e. the drug can take the role of perpetrator, victim, or even both at the same time. This definition of perpetrator and victim is to our best knowledge new in the context of PD interactions, yet similar to the use of such terms in the context of pharmacokinetic (PK) interactions, where e.g. drug elimination of a victim drug is reduced by a perpetrator⁹. To include PD interactions, we extended the sigmoidal E_{max} model of the victim drug by a perpetrator sigmoidal E_{max} term ('GPDI term') to capture the interaction effect on the level of the PD parameters. This concept generalizes the idea behind receptor-based interaction as suggested by Ariëns¹⁰. ~~The interactions in the GPDI approach are bi-directionally quantified, i.e. the drug can take the role of perpetrator, victim, or even both at the same time.~~

I. 328-334:

The GPDI approach revealed that ~~a large proportion~~ the majority of up to ~~38-67~~ % of the interactions in this dataset were of asymmetric nature where distinct perpetrator and victim information in the PD interactions was quantified, i.e. monodirectional synergism, monodirectional antagonism and asymmetric bidirectional interactions ~~the interaction could be both where synergistic synergism or antagonistic antagonism which was dependent depends~~ on the concentration ratio of the drugs.

Reviewer #2:- The four parameters for the GPDI model is not well described. Authors may want to write the GPDI values on the simulations in Figure 1.

We substantially expanded the explanation of the GPDI parameters and agree it is crucial that the parameters are clearly explained beyond the mathematical presentation. We added a supplementary Table S1, in which all GPDI parameters used for creating the scenarios presented in Figure 1 are summarized.

Changes to the manuscript:

I.102-119:

This concept generalizes the idea behind receptor-based interaction as suggested by Ariëns¹⁰. ~~The interactions in the GPDI approach are bi-directionally quantified, i.e. the drug can take the role of perpetrator, victim, or even both at the same time.~~ The PD interaction is parameterized

by INT (maximum fractional change of the victims PD parameter caused by the perpetrator with) $INT=0$: no interaction, $-1 < INT < 0$: synergy, $INT > 0$: antagonism), $EC50_{INT}$ (interaction potency) and H_{INT} (interaction sigmoidicity).

For example, for two drugs A and B with a competitive-type interaction ($EC50$ -level), the drug effects E_A and E_B are given by

$$E_A = \frac{Emax_A \times C_A^{H_A}}{\left(EC50_A \times \left(1 + \frac{INT_{AB} \times C_B^{H_{INT,B}}}{EC50_{INT,AB}^{H_{INT,B}} + C_B^{H_{INT,B}}} \right) \right)^{H_A} + C_A^{H_A}} \quad \text{Eq. 1}$$

$$E_B = \frac{Emax_B \times C_B^{H_B}}{\left(EC50_B \times \left(1 + \frac{INT_{BA} \times C_A^{H_{INT,A}}}{EC50_{INT,BA}^{H_{INT,A}} + C_A^{H_{INT,A}}} \right) \right)^{H_B} + C_B^{H_B}} \quad \text{Eq. 2}$$

INT_{AB} represents the maximum fractional change of the $EC50$ of drug A (victim) caused by drug B (perpetrator), and vice versa for INT_{BA} . $INT=0$ indicates no interaction, $-1 < INT < 0$ indicates a decrease of the $EC50$ and $INT > 0$ indicates an increase of the $EC50$. If both INT values are negative, synergy, and vice versa, antagonism is observed on the effect level. INT values of different polarities indicate an asymmetric interaction with concentration-dependent synergy and/or antagonism. In addition, potentiation (inactive drug potentiates an active drug) or coalism (inactive drugs solely jointly active) can be modelled by the GPDI approach. Implementation of the GPDI model on $EC50$ leads to a competitive interaction behavior. An interaction of allosteric type is considered when the GPDI model is implemented on E_{max} . Note that the polarity of INT is opposite when implemented on E_{max} instead of $EC50$.

Reviewer #2: Line 40: Authors claim other methods are not “conclusively interpretable”, yet Fractional Inhibitory Concentration has a simple interpretation. Also, after reading the manuscript, I am unsure if GPDI is conclusively interpretable. Authors should either clarify the model parameter interpretations or tune down their claim of interpretability.

This statement (ii) has to be seen in conjunction with statement (i). In (i) it is outlined that FIC indices are useful, but not applicable in situations where an interaction is concentration-dependent or in case of a partial agonist. In such situations, response surface analyses can be useful to explore concentration-dependencies, but no FIC indices can be derived in such situations.

We linked the two statements more closely to avoid misunderstanding.

Changes to the manuscript:

I.44-53:

(i) graphical approaches such as the isobologram method¹⁶, fractional inhibitory concentration (FIC) indices¹⁷ or the combination index¹⁸ are conceptually straightforward and useful, but their results are difficult to interpret when interactions are concentration-dependent or

isoboles are ‘curvilinear’¹⁹, e.g, when a combination partner is a partial agonist. (ii) Response surface approaches²⁰ ~~are a frequently employed in such situations that compare the observed response to a model-predicted additive response as they can~~ elucidate concentration dependencies in the interaction space, but, ~~as outlined above, are not conclusively interpretable by means of calculation and cannot be used for calculating an~~ interaction score or parameter such as FIC indices.

The reviewer is correct: The FIC indices are interpretable, but not in the same way as the GPDl parameters. GPDl parameters have a quantitative interpretation as they quantify shifts in EC50 and/or Emax. FIC indices and also e.g. the alpha parameter of the Greco model is “just a number”, i.e. cannot be interpreted, but of course compared. We have also made this aspect more clear in the introduction section.

Changes to the manuscript:

I.54-59:

(iii) FIC indices, but also Modelmodel-based approaches with a single interaction parameter^{21,22} provide interaction scores for statistical interaction assessment and can be compared, but the single point estimate of the interaction parameter has no quantitative interpretation and might not mirror the complexity of response surfaces; model-based approaches with more interaction parameters e.g. polynomials to describe the interaction surface²³, might be more flexible to fit to the data, but their interaction polynomials are also not interpretable.

Another property of the GPDl approach is that asymmetric interaction can be captured with a parametric model. None of the scoring methods is to our knowledge capable if this. We added an example for this type of interaction in the dataset, also as Figure S4.

Changes to the manuscript:

I.278-282:

24 interactions were bidirectional ~~and of opposite polarity~~ asymmetric interactions, e.g. drug Ben was antagonistic on drug Tac, but Tac potentiated the effect of Ben and ~~it will depend on the concentration whether antagonism or synergy is observed~~ concentration-dependent antagonism or synergy is observed (Figure S4).

Reviewer #2: Line 49,50: Point vi is unclear. How can drug development process effect a drug interaction?

In statement (vi) we say: “Most approaches [...] cannot be adapted to the various complexity of information obtained along the drug development process.” Therapeutic areas with a high prevalence of combination treatment such as tuberculosis research involves combination testing during drug development. The GPDl approach can adapt to various complexity of data, i.e. by using a single interaction parameter (INT=INT_AB=INT_BA) or up to four interaction parameters (INT_AB, INT_BA and both interaction EC50s). This is not the case for any other current method.

We slightly expanded to make this statement easier to grasp.

Changes to the manuscript:

I.61-63:

(vi) cannot be adapted to the various complexity of information obtained along the drug development process, i.e. reduced or more complex nested models of the same type can be applied.

Reviewer #2: Line 51: Perpetrator and victim is not yet defined.

The reviewer is correct. We decided to put this introductory statement more vaguely as a detailed definition of perpetrators and victims follows.

Changes to the manuscript:

I.63-64:

Finally, (vii) we aimed to ~~identify perpetrator and victim~~ explore the roles of each drug in PD interaction studies

Reviewer #2: Line 57: Perpetrator and victim is not mentioned in Yeh 2006.

This is correct. We are aware that there are no perpetrators and victims in Pamela Yeh's paper, but we intended to say that it might be beneficial to elucidate these roles in such networks and took this seminal paper as an example for an interaction network. We omitted the reference to avoid misunderstanding.

Reviewer #2: Line 82: competitive-type interaction not defined.

The definition was added.

Changes to the manuscript:

I.117-119:

Implementation of the GPDI model on EC_{50} leads to a competitive interaction behavior. An interaction of allosteric type is considered when the GPDI model is implemented on E_{max} .

Reviewer #2: Line 124: How is the significance calculated?

We utilized the likelihood ratio test for determining the significance of a parameter. The detailed procedure is given in the methods section in the paragraph "Parameterisation of the GPDI model from experimental data". We added the test type to the results section.

Changes to the manuscript:

I.158-161:

Model building started with a reduced GPDI model with a single interaction parameter INT and was extended to the full four parameter GPDI model, which was significant ($\alpha=0.05$, likelihood ratio test) in 152/200 scenarios (Loewe Additivity) or 167/200 (Bliss Independence).

Reviewer #2: Line 129: GPDI is a four-parameter model, while these models have one parameter. Is this comparison fair?

The Greco model and the empiric Bliss model are amongst the most frequently used PD interaction models in the literature and this is why we chose to compare against them. We however agree that the difference in number of parameters should be mentioned here. Yet, the number of parameters in the GPDI approach can range from 1 to 4 depending on the data situation and parameter significance. In addition, the GPDI parameters have a quantitative interpretation. Hence, we rather think the higher number of parameters can be seen as an advantage and the graphical fits and AIC values highly support the additional parameters. The intention was to show that many aspects are apparently missed when the simple models are used.

Changes to the manuscript:

l.175-178:

It should be noted that the conventional models solely have a single interaction parameter while the GPDI approach has up to four, but the high rate of statistical significance of the four parameter GPDI model in conjunction with the lower AIC values indicate that the additional parameters are highly supported.

Line 161-163: Although their values are different, authors can compare them using rank based methods.

The reviewer is correct. Interaction parameters of e.g. the Greco or Bliss model can be ranked, but the values are dimensionless, i.e. they cannot be interpreted. The parameters of the GPDI model instead can be quantitatively interpreted (shift of E_{max} and/or EC_{50} , interaction potency).

We clarified this in the manuscript:

l.199-201:

Yet, it should be noted that the value of α and β in the conventional models can be ranked each, but their values has-have no quantitative interpretation and hence cannot be directly compared.

Reviewer #2: Line 172-175: It seems unfair that authors claim Cokol 2011 misclassified these interactions, since that study did not look for asymmetric interactions. A better comparison might be Cokol 2014, which reported suppressive interactions using the same data set as the authors use.

This is correct and we did not at all intent to do an unfair comparison, yet aimed to compare the performance of the typically used parametric interaction models for synergy and antagonism with the newly developed GPDI model. It should be noted that the GPDI model was not used in this study to detect suppressive interactions, but to introduce (i) the concept of perpetrator and victim behavior of drugs in the context of directional PD drug interactions on the level of EC_{50} and/or E_{max} , and (ii) a parametric interaction model with interpretable parameters with high predictive performance.

Yet, the comparison with the Cokol 2014 analysis is indeed valuable regarding directions in a PD interaction. For this comparison, we specifically evaluated the suppressive interactions determined in Cokol 2014. In Cokol 2014, the authors manually determined the direction of a suppressive antagonistic interaction by assuming the less potent drug antagonizes the more potent drug. We compared these manually determined directions with the more objective model-based perpetrator-victim information obtained from the GPDI approach: In 66% of the scenarios, the GPDI analysis agreed with the directions (monodirectional antagonism and bidirectional antagonism) drawn in Cokol 2014. Different directions were identified in 20% of the cases. This indicates that the assumption that the less potent drug is the causative agent of suppression is not always true. We summarized this supplementary analysis in Table S2 and added a reference to it in the paper.

Changes to the manuscript:

I. 256-266:

A re-analysis of the Cokol dataset²⁴ revealed 61 suppressive interaction, i.e. strong antagonistic interactions with responses below one of either or both single agents. While a screen for suppressive interactions on the effect level using the GPDI approach is similar to the approach chosen by Cokol²⁴ (comparison of the combined effects with the single drug effects), the GPDI approach can enhance suppressive interactions with model-based estimates of perpetrator-victim information guided by the INT parameter. 66% of manually derived directions (assuming that the less potent drug antagonizes the more potent drug in these suppressive interactions²⁴) were in agreement with the GPDI analysis, while in the remaining cases different and/or additional directions of the PD interactions were quantified (Table S2). This indicates that the assumption that the less potent drug is the causative agent of suppression on the effect level is often, but not always true.

Reviewer #2: Line 194: Authors should define exclusive joining.

We now explicate the definition of an exclusive join. We also added the unjoined networks for the Loewe Additivity-based and Bliss Independence-based GPDI models as Figure S3 as a supplement.

Changes to the manuscript:

I.269-272:

In order not to bias ourselves towards a single additivity criterion, we exclusively joined the result of the Loewe-Additivity and Bliss-Independence-based GPDI models for the network analysis, i.e. included solely interactions that were significant under both Loewe-Additivity and Bliss-Independence, which is displayed in Figure 4a5a.

Reviewer #2: Line 219: I don't believe this is intuitive. Manuscript would benefit a lot from examples for intuition.

This is helpful, thank you. We added an example illustrating the intuitive interpretability of the GPDI model parameters.

Changes to the manuscript:

I.321-325:

For example, as illustrated in Figure 2 a and b, INT_{AB} was determined to -0.02, i.e the EC50 of drug A (Bro, bromopyruvate) was only marginally affected. Instead, INT_{BA} was determined to 15.92, i.e. in combination the EC50 of drug B (Sta, staurosporine) was increased to 1692% $((1+15.92)*100\%)$ at the EC50 of Bro. It is obvious, that the parameter α and β of the conventional models (Figure 2 c and d) cannot be quantitatively interpreted in a similar way.

Reviewer #2: - How do the INT values and interaction scores from other methods compare? Authors should include this as scatter plots.

Thank you for this suggestion. We newly created Figure 4 that visualizes the parameter values of all employed models. For easier communication, the INT values were transformed to %change of each EC50 in the combination. Figure 4 a and b indicate that the parameters alpha and beta are correlated with the EC50 shift, i.e. the INT parameters of the GPDl model.

Changes to the manuscript:

l.254-255:

The parameter values of the GPDl models as well as for the alternative models are visualized and compared in Figure 4.

Reviewer #2: Line 250-252: Authors previously claimed that these methods give similar answers.

Thank you for highlighting this. We did not intent to communicate that both methods give similar results. In fact, we write that both model describe the experimental data similarly well:

cf. l. 134-138 (original submission):

“Hence, the Loewe Additivity- and Bliss Independence-based GPDl models described the experimental data similarly well and were superior over the Greco model and the empiric Bliss Independence model which provided median AIC of -26989.4 (-35033.9; -17586.3) and -26152.5 (-33856.2; -18616.7), respectively.”

Yet, similar predictive performance does not mean that the conclusions drawn from the estimated parameters are same. We feel, that we stated very clearly in the results section that e.g. the synergy rate was higher in the Bliss-Independence-based GPDl model compared to the Loewe-GPDl-model. (cf. l. 164-191 in the original submission).

Still, we agree it might be helpful to mention the similar predictive performance in the discussion section again to avoid misunderstandings:

Changes to the manuscript:

l.366-369:

Our analysis of 200 combination experiments displayed that using Loewe Additivity or Bliss Independence can lead to different conclusions regarding synergy or antagonism on the observed interaction, yet by providing similar predictive performance.

Reviewer #2: - The arrows in Figure 4a are too small. How are these arrows interpreted?

The arrows were slightly magnified to increase readability. The arrows represent the direction of the PD interaction from perpetrator to victim. The Figure captions was expanded to put this more clearly:

I.704-710:

Figure 54: Interaction network between the visualising significant interactions from the exclusively joined Loewe Additivity and Bliss Independence-based GPDI analyses (a); arrows visualize directed interactions direction of the PD interaction from the perpetrator drug to the victim drug, i.e. decrease of the victims EC50 resulting in antagonism (red) or increase of the victims EC50 resulting in synergism (green) ~~originating from the perpetrator to the victim drug~~; note that interactions can be mono-directional or bi-directional of same polarity (joint antagonism or joint synergy) or opposite polarity leading to asymmetric bidirectional interactions;

Reviewer #2: - Can the authors comment on why some drugs show perpetrator or victim behavior?

Many of the drugs in the Cokol dataset are experimental in nature and for some their targets are unknown, hence this with this in mind this question is a difficult one. However, we agree it might be helpful to this manuscript to include some examples for perpetrator and victim behavior for drugs in the dataset where the mechanisms of action are already more clearly elucidated. We added examples for the perpetrator behaviours of bromopyruvate (monodirectional antagonisms), terfenadine (mostly monodirectional synergies), and a monodirectional antagonism between terbinafine and amphotericin B, each with an attempt of a mechanistic interpretation in relation to the GPDI model parameters.

Changes to the manuscript:

I.282-311:

Another example is the drug Ter, which is part of a larger number of synergies. It has been speculated that Ter mediates synergy through its cell-wall disrupting effect which in turn might enhance the uptake of other drugs¹⁵. When comparing the GPDI parameters of interactions where Ter was involved, diverse interactions were observed (Table S3). Yet, in the majority of the scenarios (5/11), Ter decreased the EC50 or did not significantly affect the combination partners (4/11), which might corroborate the proposed mechanism, but also indicates that further yet unknown processes are affected which also alter the EC50 of Tac. Yet, through its effect in the ergosterol pathway, Ter can also mediate monodirectional antagonism, e.g. as perpetrator on AmB, for which the INT value of 3.32 suggested an increase in EC50 to 432% $((1 + 3.32) * 100\%)$ at the EC50 of Ter, while the EC50 of Ter was not significantly altered. AmB binds ergosterol in the cell-wall; it has been proposed that AmB might lose its target when ergosterol synthesis is inhibited²⁵, which would be in agreement with the observed INT values and the identified perpetrator role of Ter from the GPDI analysis. Of the mono-directional interactions, drugs Bro, Cis, Hyg, Met and Qnn were sole perpetrators being all antagonistic on their combination partners, but never took the victim role. For Bro, there is evidence that the antagonistic interactions observed with Bro are mediated by its acidity unrelated to its precise mode of action²⁴. This distinct pattern is also found in the GPDI parameters: Bro increased the EC50 of its combination partners in 9/10 cases as perpetrator, while it was never affected itself, i.e. Bro mediated solely monodirectional antagonistic interactions (Table S4). This indicates that the effects mediated by Bro itself might not be affected by the combination partners. Drugs

Qmy, Tun, Rap, Myr and C3P were sole victim drugs, but never took the perpetrator role. Drug Tac was the most prevalent perpetrator for potentiation in 11 interactions, but 5 of those were antagonistically ‘counteracted’ by the victim. Drug Tac was also the most prevalent victim for antagonism in 9 interactions. For a large number of these interactions in the Cokol dataset, the underlying mechanism of interaction remains to be elucidated. Further research is required to elucidate the molecular interaction mechanisms to which the GPDI model might contribute by adding quantitative measures of EC50 shifts and perpetrator-victim information in order to Hence, these examples highlights that the GPDI approach provides a direction to PD interaction studies, which was used to profile the behavior of the drugs in the interaction networks (Figure 4-5 b-e).

Finally, we also updated the abstract to include more information and applied the refined definitions of asymmetric interactions as outlined above.

Changes to the manuscript:

l.20-34:

Assessment of pharmacodynamic (PD) drug interactions is a cornerstone in the development of combination drug therapies. To guide this venture, we derived a novel general pharmacodynamic interaction (GPDI) model, for ≥ 2 interacting drugs, being compatible with ~~all~~ common additivity criteria. We propose a PD interaction to be quantifiable as multidirectional shifts in drug efficacy or potency and explicate the drugs’ role as victim, perpetrator or even both at the same time. We evaluated the GPDI model against conventional approaches in a dataset of 200 combination experiments in *S. cerevisiae*: 22% interacted additively, a minority of the interactions (11%) was bidirectional antagonistic or synergistic, whereas the majority (67%) was unidirectional, i.e. asymmetric with distinct perpetrators and victims, 38% of the combinations displayed simultaneous perpetrator-victim behavior with concentration dependent synergy and antagonism, i.e. asymmetric interactions—a new class of PD interactions which is ~~not at all~~ classifiable by conventional methods. Thereby, the GPDI model excellently reflected the observed interaction data, and hence represents an attractive approach for quantitative assessment of novel combination therapies along the drug development process.

References:

1. Jaynes, J., Ding, X., Xu, H., Wong, W. K. & Ho, C. M. Application of fractional factorial designs to study drug combinations. *Stat. Med.* **32**, 307–318 (2013).
2. Wood, K., Nishida, S., Sontag, E. D. & Cluzel, P. Mechanism-independent method for predicting response to multidrug combinations in bacteria. *Proc. Natl. Acad. Sci.* **109**, 12254–12259 (2012).
3. Segrè, D., Deluna, A., Church, G. M. & Kishony, R. Modular epistasis in yeast metabolism. *Nat. Genet.* **37**, 77–83 (2005).
4. Beppler, C. *et al.* Uncovering emergent interactions in three-way combinations of stressors. *J. R. Soc. Interface* **13**, (2016).
5. Tekin, E. *et al.* Enhanced identification of synergistic and antagonistic emergent interactions among three or more drugs. *J. R. Soc. Interface* **13**, 18–20 (2016).

6. Chevereau, G. & Bollenbach, T. Systematic discovery of drug interaction mechanisms. *Mol. Syst. Biol.* **11**, 807–807 (2015).
7. Hill, A. V. The possible effects of the aggregations of the molecules of haemoglobin on its dissociation curves. *J. Physiol.* **40**, 4–7 (1910).
8. Michaelis, L. & Menten, M. Die Kinetik der Invertinwirkung. *Biochem. Z.* **49**, 333–369 (1913).
9. Prueksaritanont, T. *et al.* Drug – Drug Interaction Studies : Regulatory Guidance and An Industry Perspective. *AAPS J.* **15**, 629–645 (2013).
10. Ariens, E. J., Van Rossum, J. M. & Simonis, A. M. Affinity, intrinsic activity and drug interactions. *Pharmacol. Rev.* **9**, 218–36 (1957).
11. Zimmer, A., Katzir, I., Dekel, E., Mayo, A. E. & Alon, U. Prediction of multidimensional drug dose responses based on measurements of drug pairs. *Proc. Natl. Acad. Sci.* 201606301 (2016). doi:10.1073/pnas.1606301113
12. Beppler, C. *et al.* When more is less: Emergent suppressive interactions in three-drug combinations. *BMC Microbiol.* **17**, 107 (2017).
13. Clewe, O., Wicha, S. G., de Vogel, C., de Steenwinkel, J. E. M. & Simonsson, U. S. H. A model-informed pre-clinical approach for prediction of clinical pharmacodynamic interactions of anti-tuberculosis drug combinations. *J. Antimicrob. Chemother.* **submitted**, (2017).
14. Chen, C. *et al.* Assessing Pharmacodynamic Interactions in Mice using the Multistate Tuberculosis Pharmacometric and General Pharmacodynamic Interaction Models. *CPT Pharmacometrics Syst. Pharmacol.* **accepted**, (2017).
15. Cokol, M. *et al.* Systematic exploration of synergistic drug pairs. *Mol. Syst. Biol.* **7**, 544 (2011).
16. Tallarida, R. J. An overview of drug combination analysis with isobolograms. *J. Pharmacol. Exp. Ther.* **319**, 1–7 (2006).
17. Odds, F. C. Synergy, antagonism, and what the chequerboard puts between them. *J. Antimicrob. Chemother.* **52**, 1 (2003).
18. Chou, T. C. Drug combination studies and their synergy quantification using the chou-talalay method. *Cancer Res.* **70**, 440–446 (2010).
19. Grabovsky, Y. & Tallarida, R. J. Isobolographic analysis for combinations of a full and partial agonist: curved isoboles. *J. Pharmacol. Exp. Ther.* **310**, 981–986 (2004).
20. Wicha, S. G., Kees, M. G., Kuss, J. & Kloft, C. Pharmacodynamic and response surface analysis of linezolid or vancomycin combined with meropenem against *Staphylococcus aureus*. *Pharm. Res.* **32**, 2410–2418 (2015).
21. Greco, W. R., Bravo, G. & Parsons, J. C. The search for synergy: a critical review from a response surface perspective. *Pharmacol. Rev.* **47**, 331–85 (1995).
22. Lee, J. J., Kong, M., Ayers, G. D. & Lotan, R. Interaction index and different methods for determining drug interaction in combination therapy. *J. Biopharm. Stat.* **17**, 461–480 (2007).
23. Minto, C. F. *et al.* Response surface model for anesthetic drug interactions. *Anesthesiology* **92**, 1603–1616 (2000).
24. Cokol, M. *et al.* Large-scale identification and analysis of suppressive drug interactions. *Chem. Biol.* **21**, 541–551 (2014).
25. Johnson, M. D., Macdougall, C., Ostrosky-zeichner, L., Perfect, J. R. & Rex, J. H. Combination

Antifungal Therapy. *Antimicrob Agents Chemother* **48**, 693–715 (2004).

Reviewers' comments:

Reviewer #1 (Remarks to the Author):

I felt the author's did a good job addressing all of the reviewers' concerns. I looked at not only my comments but the others reviewer's comments and felt the revisions were extremely thorough and thoughtful.

Reviewer #2 (Remarks to the Author):

The revised manuscript by Wicha et al clarifies some of the points I raised earlier. Here, the aims and components of their GPDI model is better explained and a more comprehensive comparison to previous approaches is given. I believe this is an exciting study, and the model described herein sheds new light on the analysis of drug interactions. Their model is different from previous models because it may have up to four parameters instead of one parameter in previous approaches. Extra parameters allow new interpretations for the interaction between drugs, which is exciting and could be helpful for learning new biology.

However I am concerned by the language and some claimed benefits of the model. That a 4-parameter model better explains a data than a 1-parameter model, does not make the previous model wrong. Similarly, if someone publishes a 10-parameter model next year, this would not make the GPDI model wrong. I believe authors should carefully clarify what they have achieved. Their finding and model is exciting, but it is not the first drug interaction model with an interpretation, it does not predict and previous methods have also been used for simulation. Making far reaching conclusions and unconvincing claims about their methodology detracts from their actual findings. My concerns are addressable with minor revisions using only language edits and require no additional analysis.

Major concerns:

1. I have previously commented that GPDI model does not predict, and that authors should make this clear. In their response authors agreed that their model does not predict using other external experimental data, but they say that GPDI model predicts observed PD interactions "spot-on". Which PD interactions? If PD interactions are interaction parameters reported in previous studies, then this is not prediction as it would just mean that authors' parameters and previous measurements agree. Prediction only pertains to making guesses about external data (prospective validation, unseen data or in the case of cross-validation, held-out data). I feel strongly that authors never predict and thus should refrain from the use of this word or explain what they predict using what data.

2. I have previously criticized the authors' claim of interpretability of previous methods. The response of the authors is far from clear. I understand that previous methods cannot measure asymmetric interactions, but they do give quantitative results, they can be ranked and they can be (has been) interpreted as synergy or antagonism. As a simple example, an FIC score of 0.5 means that half total dose of a combination gives same phenotype as full dose of constituent drugs. But the authors directly contradict this well known fact in lines 54-56. In another example, I cannot understand what the authors mean by lines 200-201. "Yet, it should be noted that the value of α and β in the conventional models can be ranked each, but their values has have no quantitative interpretation and hence cannot be directly compared." As far as I understand, if something can be ranked, then it is quantitative, then they can de directly compared. In addition, authors do compare alpha and beta values in Figure 3c, contradicting this statement. FIC, alpha and beta values are just "numbers"; these numbers are generated by models with well defined

interpretations.

3. Authors claim in line 53 and 377 that, previous methodologies are not exploitable for computer simulation. Why not? They are numerical models and can be simulated, also they have been heavily used for drug interaction simulations (for example, see: <https://www.ncbi.nlm.nih.gov/pubmed/17332758>)

Minor concerns:

1. The newly included GPDI parameters in Table S1 are highly instructive. It would be very useful if these parameters were included in Figure 1.
2. Authors find protagonists in drug interactions, not drug interaction studies. Title should be changed accordingly.
3. Table S3, terbinfain
4. Line 289, authors write Tac and this is out of context, do they mean Ter?
5. Line 355, refers to two studies using GPDI model. These studies are unpublished.
6. Line 489, authors may want to clarify why log₂ drug dilutions are used for simulation, while linear drug doses were used in the rest of manuscript and the analyzed 200 experiments.
7. Authors use EC₅₀ for their entire analysis, however there is nothing special about EC₅₀. A good test for the robustness of GPDI model might be using the model for several different EC levels and showing that results do not depend on EC level chosen.
8. What are alpha and beta shown in Figure 2 and 3? How are these computed and are they in agreement with the loewe and bliss based scores in 200 experiments? The agreement of greco and bliss models is well-known and need not be shown as a figure (Figure 3c).

Point-by-point response to the revision of the manuscript:

On perpetrators and victims: A general pharmacodynamic interaction model identifies the protagonists in drug interactions.

Sebastian G. Wicha, Chunli Chen, Oskar Clewe and Ulrika S.H. Simonsson

Nature Communications, NCOMMS-16-26282-T

Line numbers of the reviewers refer to the original submission, line numbers of our responses and changes to the manuscript refer to the revised version of the manuscript incl. track-changes.

Reviewers' comments:

Reviewer #1 (Remarks to the Author):

I felt the author's did a good job addressing all of the reviewers' concerns. I looked at not only my comments but the others reviewer's comments and felt the revisions were extremely thorough and thoughtful.

Response: Thank you! We appreciate your positive feedback.

Reviewer #2 (Remarks to the Author):

The revised manuscript by Wicha et al clarifies some of the points I raised earlier. Here, the aims and components of their GPDI model is better explained and a more comprehensive comparison to previous approaches is given. I believe this is an exciting study, and the model described herein sheds new light on the analysis of drug interactions. Their model is different from previous models because it may have up to four parameters instead of one parameter in previous approaches. Extra parameters allow new interpretations for the interaction between drugs, which is exciting and could be helpful for learning new biology.

Response: Thank you for your positive feedback. We are positive, that we will be able to clarify the remaining aspects you raised as well (see below).

However I am concerned by the language and some claimed benefits of the model. That a 4-parameter model better explains a data than a 1-parameter model, does not make the previous model wrong. Similarly, if someone publishes a 10-parameter model next year, this would not make the GPDI model wrong. I believe authors should carefully clarify what they have achieved. Their finding and model is exciting, but it is not the first drug interaction model with an interpretation, it does not predict and previous methods have also been used for simulation. Making far reaching conclusions and unconvincing claims about their methodology detracts from their actual findings. My concerns are addressable with minor revisions using only language edits and require no additional analysis.

Response: Thank you for this constructive feedback. We respond to each specific point below.

Major concerns:

1. I have previously commented that GPDI model does not predict, and that authors should make this

clear. In their response authors agreed that their model does not predict using other external experimental data, but they say that GPDI model predicts observed PD interactions “spot-on”. Which PD interactions? If PD interactions are interaction parameters reported in previous studies, then this is not prediction as it would just mean that authors’ parameters and previous measurements agree. Prediction only pertains to making guesses about external data (prospective validation, unseen data or in the case of cross-validation, held-out data). I feel strongly that authors never predict and thus should refrain from the use of this word or explain what they predict using what data.

Response:

We feel this is a misunderstanding between the author’s and the reviewer on what a prediction is. In the pharmacometric literature, it is widely accepted that a prediction is a simulation from a model, regardless of the nature of the parameters, i.e. estimated from the data or not.

Yet, we see the problem of misunderstanding the term prediction and appreciate that the reviewer highlights this. Hence, we will change the wording from “prediction” to “description” of the observed PD interaction (which are the time courses of the combined drug effects).

Changes to the manuscript:

Eliminated “predict” or “prediction” and changed to “describe” or “description” or explicated what is meant in lines 77, 160, 161, 163, 171, 299, 353, 493, 545, 681 and 684.

2. I have previously criticized the authors’ claim of interpretability of previous methods. The response of the authors is far from clear. I understand that previous methods cannot measure asymmetric interactions, but they do give quantitative results, they can be ranked and they can be (has been) interpreted as synergy or antagonism. As a simple example, an FIC score of 0.5 means that half total dose of a combination gives same phenotype as full dose of constituent drugs. But the authors directly contradict this well known fact in lines 54-56.

Response:

The reviewer is correct – FIC indices are interpretable. Yet, lines 54 to 56 were intended to refer to the cited polynomial models (Minto et al.), which has parameters that are itself not interpretable. We apologize, that we have been imprecise and wrongly included FIC indices into the statement. We have corrected the manuscript accordingly.

Changes to the manuscript (l. 52-58):

(iii) FIC indices, but also model-based approaches with a single interaction parameter^{10,11} provide interaction scores for statistical interaction assessment and can be compared, but the single point estimate ~~of the interaction parameter has no quantitative interpretation and~~ might not mirror the complexity of response surfaces; model-based approaches with more interaction parameters e.g. polynomials to describe the interaction surface¹², might be more flexible to fit to the data, but their interaction polynomials are ~~also~~ not interpretable.

In another example, I cannot understand what the authors mean by lines 200-201. “Yet, it should be noted that the value of α and β in the conventional models can be ranked each, but their values has have no quantitative interpretation and hence cannot be directly compared.” As far as I understand, if something can be ranked, then it is quantitative, then they can be directly compared. In addition,

authors do compare alpha and beta values in Figure 3c, contradicting this statement. FIC, alpha and beta values are just “numbers”; these numbers are generated by models with well defined interpretations.

Response:

It is correct that the parameters from the other models are quantitative measures and can be ranked and compared using the same model, i.e. one can compare a scenario A with a scenario B using the Greco model. Yet, a comparison in a scenario A comparing the Greco model with the Bliss model is more difficult in our opinion, which is the point we intended to make. For example, it is unclear what an alpha parameter (Loewe Additivity-based Greco model) of 0.25 means in relation to a beta parameter of 0.25 (Empiric Bliss model). In contrast to that, the GPDI model always quantifies the PD interaction as shift of EC50 and Emax. Hence a direct and interpretable comparison between the Loewe Additivity GPDI and Bliss Independence GPDI models can be made.

However, we agree that our phrasing of this is somewhat cryptic and, so we tried to explicate the example above and eliminated the statement that caused this misunderstanding, as it is not central to our study.

Changes to the manuscript (l. 197-199, 366-372):

~~Yet, it should be noted that the value of α and β in the conventional models can be ranked each, but their values have no quantitative interpretation and hence cannot be directly compared.~~

Moreover, the GPDI approach comes with further numerous advantages over existing approaches, which comprise quantitatively interpretable interaction point estimates across Loewe Additivity and Bliss Independence, no requirement of prior knowledge on the precise mode of (inter-)action, flexibility to adapt to multi-drug combination data of various complexity, compatibility with established additivity criteria, provision of insight into perpetrators and victims in PD interaction networks, and the possibility to predict describe time-courses of the interaction.

3. Authors claim in line 53 and 377 that, previous methodologies are not exploitable for computer simulation. Why not? They are numerical models and can be simulated, also they have been heavily used for drug interaction simulations (for example, see: <https://www.ncbi.nlm.nih.gov/pubmed/17332758>)

Response:

The reviewer is right. This statement is referring to a pure response surface analysis that solely relates the observed combined response to a hypothetical additive response generated by a Null interaction / Additivity model. No parameters are estimated there and hence the model cannot be used to simulate time-courses of the combined effects. We agree that this is not true in general, e.g. for FIC indices, although the simulations from an FIC-based model might be biased in case of concentration-dependent PD interactions. We explicate now what we mean by “computer

simulation" (simulation of time-courses of the combined drug effect) and have corrected the statement regarding the FIC indices.

Changes to the manuscript (l. 48-52):

Both As response surface analyses represent a pure comparison between observed and additive response, aforementioned methodologies are they cannot exploitable be used for computer longitudinal simulations of the observed interaction pattern at (changing) concentrations over time (pharmacokinetic-pharmacodynamic simulations).

Minor concerns:

1. The newly included GPD1 parameters in Table S1 are highly instructive. It would be very useful if these parameters were included in Figure 1.

Response:

Thank you. We agree that your suggestion to add the parameter values was very helpful. Yet, Figure 1 becomes very messy when all parameter values are added. We therefore would like to keep it as Table S1. Another option would be to include the values as Table 1. We leave this decision to the Editor if the space permits this.

2. Authors find protagonists in drug interactions, not drug interaction studies. Title should be changed accordingly.

Response:

Thanks. We have changed the title accordingly.

3. Table S3, terbinfain

Response:

Thanks for the thorough review. We corrected the typo.

4. Line 289, authors write Tac and this is out of context, do they mean Ter?

Response:

You are right. It should read Ter – Thanks!

5. Line 355, refers to two studies using GPD1 model. These studies are unpublished.

Response:

Both studies are accepted in the meantime. We updated the references accordingly to 'in press' as these have not been assigned to an issue yet.

6. Line 489, authors may want to clarify why log2 drug dilutions are used for simulation, while linear

drug doses were used in the rest of manuscript and the analyzed 200 experiments.

Response:

This is a very good remark. We used the log₂ dilution design for this general part, as this is in our view the more commonly used design. We also evaluated a linear design where the EC₅₀ was placed in the 40-60% range of the maximum concentration studied. Concentrations were also setup in an 8x8 design. While the median RSE was a bit higher in this scenario (30% for INT values and 52% for EC_{50_INT}), all parameters were still well identifiable. We added this information to the manuscript.

Changes to the manuscript:

I.132-134:

A log₂ dilution design is preferable over a linear design. While still being identifiable, the anticipated median relative standard errors were 30.0% (11.1-268%) for INT and 52.4% (23.2-339%) for EC_{50_INT} in the linear design.

I. 490-492:

A linear dilution scheme was also assessed where the EC₅₀ values were placed in the 40-60% range of the highest concentration studied, while all other parameters were as described above.

7. Authors use EC₅₀ for their entire analysis, however there is nothing special about EC₅₀. A good test for the robustness of GPDI model might be using the model for several different EC levels and showing that results do not depend on EC level chosen.

Response:

This is an interesting idea. The EC₅₀ was used in the present study as it is together with E_{max}, the key parameter of the Michaelis-Menten model and most frequently used to determine drug potency and efficacy. It would be interesting to test the GPDI concept in other effect markers, e.g. the stationary concentration or other EC levels as well. We added this idea to the manuscript.

I. 372-374:

Future studies should also evaluate the utility of the GPDI approach at other EC levels or the stationary concentration³⁹ as potency markers.

8. What are alpha and beta shown in Figure 2 and 3? How are these computed and are they in agreement with the loewe and bliss based scores in 200 experiments? The agreement of greco and bliss models is well-known and need not be shown as a figure (Figure 3c).

Response:

The detailed definitions of the parameter alpha of the Greco model and beta of the Empiric Bliss model are given in the Methods section under the heading "Assessment of the performance of the GPDI approach and comparison to conventional methods". Figure 3c does not show a comparison between Greco and Bliss model, but between GPDI and Greco model. We think the reviewer wants to point towards Figure 4c where the alpha and beta parameters are presented. We think it is a

useful plot as it shows an agreement, but also highlights the numeric differences between both model parameters. We hence would like to keep it, but would leave to the Editor to decide whether there is enough space for it.

REVIEWERS' COMMENTS:

Reviewer #2 (Remarks to the Author):

The authors answered all my criticisms and have clarified the aspects that were cryptic in initial submission. I recommend it for publication.